# A Closer Look at Transformers for Time Series Forecasting: Understanding Why They Work and Where They Struggle

**Yu Chen** [1 2]   **Nathalia Céspedes** [1 2]   **Payam Barnaghi** [1 2]

## Abstract

Time-series forecasting is crucial across various domains, including finance, healthcare, and energy. Transformer models, originally developed for natural language processing, have demonstrated significant potential in addressing challenges associated with time-series data. These models utilize different tokenization strategies, point-wise, patch-wise, and variate-wise, to represent time-series data, each resulting in different scope of attention maps. Despite the emergence of sophisticated architectures, simpler transformers consistently outperform their more complex counterparts in widely used benchmarks. This study examines why point-wise transformers are generally less effective, why intra- and inter-variate attention mechanisms yield similar outcomes, and which architectural components drive the success of simpler models. By analyzing mutual information and evaluating models on synthetic datasets, we demonstrate that intra-variate dependencies are the primary contributors to prediction performance on benchmarks, while inter-variate dependencies have a minor impact. Additionally, techniques such as Z-score normalization and skip connections are also crucial. However, these results are largely influenced by the self-dependent and stationary nature of benchmark datasets. By validating our findings on real-world healthcare data, we provide insights for designing more effective transformers for practical applications.

[1]Imperial College London, Department of Brain Sciences, London, W12 0NN, United Kingdom [2]The UK Dementia Research Institute, Care Research and Technology Centre, United Kingdom. Correspondence to: Yu Chen <yu.chen@imperial.ac.uk>, Payam Barnaghi <p.barnaghi@imperial.ac.uk>.

*Proceedings of the 42$^{nd}$ International Conference on Machine Learning*, Vancouver, Canada. PMLR 267, 2025. Copyright 2025 by the author(s).

## 1. Introduction

Time-series forecasting plays a crucial role in numerous critical applications, from finance and healthcare to energy and transportation. Accurate predictions of future values based on historical data are indispensable for informed decision-making and strategic planning across these domains. Building on their success in natural language processing, transformer models (Vaswani et al., 2017) have rapidly gained attention from researchers in time-series forecasting. Compared to RNN-based models (Hochreiter & Schmidhuber, 1996; Rangapuram et al., 2018; Salinas et al., 2020), transformers are capable of capturing long-range dependencies and efficiently processing large-scale datasets, making them a promising solution for addressing common challenges in time series data.

Based on how tokens are represented in the attention mechanism, transformer-based models for time series can be categorized into three types: point-wise, patch-wise, and variate-wise (Wang et al., 2024b). In point-wise approaches, the embedding of each token is derived from the values of all variates at a specific time step. Patch-wise methods, on the other hand, represent small segments of the time series as tokens, with each token's embedding generated from a segment of a single variate over a fixed time window. variate-wise approaches take a more global perspective by treating the entire time series of a variate as a single token. A visualization of these three token types for time series data is shown in Figure 1.

Recent research on transformers for time series forecasting has explored various tokenization strategies, with many models incorporating hybrid token scopes to enhance performance. However, point-wise transformers are generally less competitive than patch-wise and variate-wise ones. Interestingly, some transformers with simpler token scopes and attention mechanisms, such as iTransformer (Liu et al., 2023) and PatchTST (Nie et al., 2023), perform exceptionally well on standard time series forecasting benchmarks (Wang et al., 2024b). Notably, iTransformer is a variate-wise transformer with inter-variate attention, where attention is computed across different variates. In contrast, PatchTST is a patch-wise transformer with intra-variate attention, focusing on interactions between patches from the

Figure 1: Demonstration of different token representations in time-series transformers. A point-wise token is formed by all variate values at one time step; a patch-wise token is formed by a segment or patch of a single variate in one time window; a variate-wise token is the entire time series of a single variate.

same variate. Despite these fundamental differences, both models often achieve similar performance in benchmark.

This study seeks to address the following key questions:

1. Why point-wise transformers generally less competitive in time series forecasting?

2. Why do transformers with intra-variate attention and those with inter-variate attention perform similarly?

3. Why do transformers with basic attention mechanisms excel in time series forecasting?

4. Which components in the basic transformer's architecture contribute most to the success in time series forecasting?

To achieve this goal, we conducted a comprehensive analysis of selected representative transformers for time series forecasting (Table 1). To ensure a unified comparison across models with varying architectures and attention mechanisms, we introduce new metrics that estimate the mutual information within and between variates based on a model's input and output, capturing both intra-variate and inter-variate information flows in a model-agnostic manner. Beyond standard benchmark datasets from the literature, we also designed a set of synthetic datasets with controlled intra- and inter-variate dependencies to systematically assess transformer performance across different conditions.

Our findings provide valuable insights for understanding how transformers work in time series forecasting:

1. We demonstrate that transformers with superior performance excel at capturing intra-variate patterns, while inter-variate patterns play a much smaller role in model predictions, even in transformers equipped with inter-variate attention mechanisms. This can be attributed to the fact that, in the majority of time series forecasting benchmarks, each variate is largely self-dependent. These findings provide valuable insights into questions 1 and 2.

2. The basic transformer encoder is capable of capturing the intra-variate dependencies when using patch-wise and variate-wise tokens due to the self-attention mechanism. The skip-connection in encoder layers plays an crucial role for learning intra-variate patterns. A variate-independent decoder also helps the model to focus on the intra-variate dependencies. These findings address the question 3 and 4.

3. A key factor that contributes to the success of models like iTransformer and PatchTST is not the model's architecture itself, but rather the use of Z-score normalization and denormalization for the model's input and output. This technique is particularly effective when the variates are stationary throughout the combined duration of the observation and prediction periods. However, it may degrade a model's forecasting performance when this assumption is violated.

These findings also help explain certain phenomena observed in the literature. (Tan et al., 2024) found that pretrained large language models are not particularly useful for time series forecasting, as understanding a variate's broader context is not essential for capturing intra-variate patterns. However, this conclusion is based on commonly used benchmarks and may not hold for fundamentally different datasets, such as event time series. Additionally, (Zeng et al., 2023) suggested that linear models can outperform certain transformers on time series forecasting benchmarks, as point-wise transformers struggle to effectively capture intra-variate patterns.

Moreover, the findings of this study offer complementary perspectives to several prior works. For instance, Zhao & Shen (2024) introduced a plugin method to identify and exploit locally stationary lead-lag relationships between variates to improve forecasting performance. Their work focuses on a specific type of inter-variate dependency and demonstrated modeling such relationships can be beneficial. Another relevant work, Reversible Instance Normalization (RevIN) (Kim et al., 2021), proposed a method similar to Z-score normalization to address distributional shifts in time series data. While we also examined Z-score normalization in this study, a key difference is that RevIN incorporates learnable parameters within its normalization process, whereas the Z-score normalization in our study is the standard, non-learnable version. Notably, we observed

that Z-score normalization degraded model performance on synthetic datasets which are non-stationary with monotonic trends, an outcome that contrasts with observations in RevIN.

## 2. Related work

We selected a set of time series transformers in our analysis, which are representative of the different token scopes and attention mechanisms used for time series forecasting. These models are summarized in Table 1 and more details are provided below:

1). Transformer (Vaswani et al., 2017) – The standard transformer architecture with point-wise tokens.

2.) Autoformer (Wu et al., 2021) – Autoformer introduces autocorrelation attention to capture long-range dependencies in time series. It decomposes the time series into trend and seasonal components using a moving average method, as in traditional time series models (Box et al., 2015). Additionally, a point-wise projection is added to each encoder layer to capture inter-variate dependencies. Since autocorrelation attention is computed over the entire time series, we consider Autoformer as a hybrid model that incorporates variate-wise and point-wise tokens.

3). FEDformer (Zhou et al., 2022) – FEDformer introduces frequency-enhanced attention to capture periodic patterns in time series data. Building on trend and seasonal decomposition, it adds frequency decomposition for seasonal components. The architecture of FEDformer closely resembles Autoformer, also incorporating a point-wise projection in its encoder layers.

4). Crossformer (Zhang & Yan, 2023) – Crossformer introduces a Two-Stage Attention mechanism to capture both intra- and inter-variate dependencies. It employs hierarchical patch-wise tokens for learning cross-time (intra-variate) and cross-dimension (inter-variate) attention efficiently.

5). PatchTST (Nie et al., 2023) – PatchTST uses a vanilla transformer encoder architecture but with patch-wise tokens. It segments the time series along the time axis into patches and applies attention to these patches for each variate. The model is incapable of capturing inter-variate patterns, as its decoder, a linear projection layer, is also variate-independent.

6). iTransformer (Liu et al., 2023) – iTransformer uses the same encoder architecture as the vanilla transformer but with variate-wise tokens. It treats the entire time series of a variate as a single token, thereby learning inter-variate attention through the multi-head self-attention mechanism. It's decoder is variate-independent linear projection, the same as PatchTST.

7). TimeXer (Wang et al., 2024a) – TimeXer combines the ideas of PatchTST and iTransformer to enhance its ability of capturing both intra- and inter-variate dependencies. It learns intra-variate attention via patch-wise tokens and inter-variate attention through variate-wise tokens. The output of its encoder layers is a concatenation of the outputs from both attention mechanisms. It's decoder is the same as PatchTST and iTransformer.

Some other point-wise approaches, such as Reformer (Kitaev et al., 2020), Pyraformer (Liu et al., 2022), and Informer (Zhou et al., 2021), were not included in our experiments. This is because they are generally perform worse and their architectures are less representative. Temporal Fusion Transformer (TFT) (Lim et al., 2021), however, is a hybrid model that incorporates variable selection and LSTM-based encoders prior to applying attention. As a result, its token representations are less explicit and not as directly comparable to the models we selected. Additionally, Tan et al. (2024) introduces PAttn, a patch-wise model similar to PatchTST in both token scope and architecture. Therefore, we consider PatchTST a representative model for PAttn. Furthermore, this study focuses on lightweight transformers for time series forecasting and does not include pretrained large language models.

Traditional statistical models such as ARIMA (AutoRegressive Integrated Moving Average) (Box & Pierce, 1970), were not considered in this study as we are focusing on transformer-based models. However, such methods can be useful for analyzing the properties of single variates and provide certain interpretability to time series forecasting.

Additionally, Qiu et al. (2024) introduced new benchmarks for time series forecasting, primarily focusing on intra-variate perspectives, which differ significantly from the scope of the synthetic datasets in our study.

## 3. Method

Standard metrics for time series forecasting, such as mean squared error (MSE) and mean absolute error (MAE), are insufficient for evaluating a model's ability to learn intra- and inter-variate patterns. To overcome this limitation, we propose new metrics that estimate the mutual information (Cover, 1999) between a model's input and output, capturing intra- and inter-variate information flows in a model-agnostic way. These metrics facilitate comparisons across models with different architectures.

Notations: Let $\mathbf{x} = \{\mathbf{x}_1, \mathbf{x}_2, \ldots, \mathbf{x}_M\}$ be a multivariate time series data sample, where $\mathbf{x}_i = \{x_{i,1}, x_{i,2}, \ldots, x_{i,T}\}$ is the $i$-th variate with $T$ time steps. The goal of time series forecasting is to predict the future values of each variate in $\mathbf{x}$ based on the historical values. We denote the predicted values as $\hat{\mathbf{x}}_i = \{\hat{x}_{i,T+1}, \hat{x}_{i,T+2}, \ldots, \hat{x}_{i,T+k}\}$. In addition,

Table 1: Selected transformer-based models for time series forecasting in our analysis. Some models are hybrid that combines multiple token scopes in the model architecture. We consider Autoformer and FEDformer as hybrid with point-wise tokens because they both include point-wise projection in their encoders. Z-norm indicates whether the model uses Z-score normalization and denormalization for the input and output.

| | Point-wise | Patch-wise | Variate-wise | Inter-variate attention | Z-norm | Variate-independent decoder |
|---|---|---|---|---|---|---|
| Transformer (Vaswani et al., 2017) | ✓ | | | ✓ | | |
| Autoformer (Wu et al., 2021) | ✓ | | ✓ | ✓ | | |
| FEDformer (Zhou et al., 2022) | ✓ | | ✓ | ✓ | | |
| Crossformer (Zhang & Yan, 2023) | | ✓ | | ✓ | | |
| PatchTST (Nie et al., 2023) | | ✓ | | | ✓ | ✓ |
| iTransformer (Liu et al., 2023) | | | ✓ | ✓ | ✓ | ✓ |
| TimeXer (Wang et al., 2024a) | | ✓ | ✓ | ✓ | ✓ | ✓ |

$\mathbf{x}_{/i} = \{\mathbf{x}_j | \forall j \neq i\}$ represents the data sample excluding the $i$-th variate.

To quantify the dependency of the model's prediction for the $j$-th variate ($\hat{\mathbf{x}}_j$) on the $i$-th variate ($\mathbf{x}_i$), we estimate the mutual information between them, conditioned on $\mathbf{x}_{/i}$:

$$I(\hat{\mathbf{x}}_j; \mathbf{x}_i | \mathbf{x}_{/i}) = H(\hat{\mathbf{x}}_j | \mathbf{x}_{/i}) - H(\hat{\mathbf{x}}_j | \mathbf{x}_i, \mathbf{x}_{/i}) \quad (1)$$

Since the model's output is deterministic given a complete input, the conditional entropy $H(\hat{\mathbf{x}}_j | \mathbf{x}_i, \mathbf{x}_{/i}) = 0$. Hence:

$$I(\hat{\mathbf{x}}_j; \mathbf{x}_i | \mathbf{x}_{/i}) = H(\hat{\mathbf{x}}_j | \mathbf{x}_{/i}) \propto \sum_{k=1}^{K} \log \sigma_{ij,k}, \quad (2)$$

Here $k$ is the $k$-th time step of $\hat{\mathbf{x}}_j$ and we assume $\hat{\mathbf{x}}_j | \mathbf{x}_{/i} \sim \mathcal{N}(\mu_{ij}, \sigma_{ij}^2 \mathbf{I})$, $\mathbf{I}$ is the identity matrix. $\boldsymbol{\sigma}_{ij}^2$ estimates the extent to which changes in the prediction of variate j are caused by changes in variate i. Mathematically, $\boldsymbol{\sigma}_{ij}^2$ is the variance of the predictions of variate j conditioned on inputs x where all variates are held fixed except for variate i. It is straightforward to see that the mutual information $I(\hat{\mathbf{x}}_j; \mathbf{x}_i | \mathbf{x}_{/i})$ can be evaluated by the standard deviation of $\hat{\mathbf{x}}_j | \mathbf{x}_{/i}$. Therefore, we define a mutual information score as below:

$$\bar{\boldsymbol{\sigma}}_{ij} = \frac{1}{N} \sum_{n=1}^{N} \frac{1}{K} \sum_{k=1}^{K} \sigma_{ij,k}^{(n)}, \quad \forall i,j \in \{1, 2, \ldots, M\} \quad (3)$$

Where $N$ is the number of samples, $K$ is the predicted sequence length. Unlike Pearson's correlation, it is more versatile as it captures both linear and non-linear relationships.

To estimate this conditional variance, the variation of variate i is introduced through N different samples by augmenting original samples in the dataset. Specifically, each original sample is augmented into N = 5 versions, differing only in the value of variate i: one instance is set to zero, one

retains the original value, and the remaining instances are generated by adding Gaussian noise of varying strengths to the original value.

When $j = i$, we call $\bar{\boldsymbol{\sigma}}_{ii}$ as the intra-variate mutual information score (Intra MI), and when $j \neq i$, we call $\bar{\boldsymbol{\sigma}}_{ij}$ as the inter-variate mutual information score (Inter MI). For a time series with more than two variates, we define two metrics for evaluating inter-variate mutual information across all variates:

(i) Average Inter MI (Avg Inter MI): The average of all inter-variate mutual information scores.

$$\frac{1}{M(M-1)} \sum_{i=1}^{M} \sum_{j=1, j \neq i}^{M} \bar{\boldsymbol{\sigma}}_{ij}$$

(ii) Maximum Inter MI (Max Inter MI): The maximum inter-variate mutual information score.

$$\max_{i,j} \bar{\boldsymbol{\sigma}}_{ij}, \quad \forall i,j \in \{1, 2, \ldots, M\}, i \neq j$$

We propose Max Inter MI as a measure of the mutual information captured by a model between the most strongly interacting variates. This metric helps assess whether a model is effectively learning inter-variate dependencies, particularly in cases where the number of variates is large and only a few exhibit strong interactions. For instance, in Figure 4, Crossformer achieves the highest Max Inter MI on the Traffic dataset (which has 862 variates), while its average Avg Inter MI remains lower than that of most other models.

## 4. Experiments

We conducted a series experiments to analyze the behavior of selected transformers on both real-world and syn-

thetic datasets. The experiments were implemented using the Time Series Library (Wang et al., 2024b). To ensure consistency, we employed a unified configuration across all prediction lengths within a given dataset for each model. As a result, the reported performance may differ from that presented in the original papers for these models. All the experimental results in this work are averaged over 3 runs with different random seeds. Standard deviation of the results are provided in the Appendix. Code are available here: `https://github.com/yc14600/TimeSeries-Transformers-Analysis`.

### 4.1. Datasets

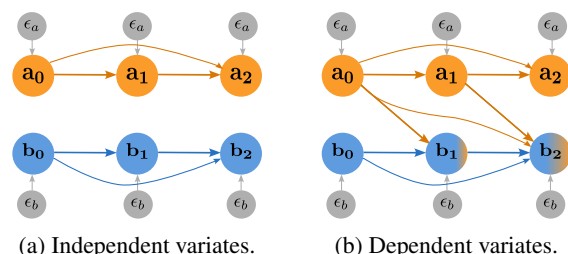

(a) Independent variates.    (b) Dependent variates.

Figure 2: Demonstration of synthetic datasets with independent and dependent variates.

We included the following datasets in our experiments, as they are among the most commonly used benchmarks in the literature: Weather, Electricity, Traffic, and ETT (comprising four subsets). More details of these datasets are provided in Table 1 in the Appendix.

Additionally, we designed a set of synthetic datasets with controlled intra- and inter-variate dependencies to evaluate the performance of different transformers under various conditions. Each synthetic dataset consists of two variates and is generated as hourly time series over a one-year period. Based on the relationship between the variates, we classify these datasets into two categories: independent and dependent (Figure 2).

For the independent variates, we generated them independently with the following process:

(1) We begin by generating a trend component characterized by a specified autocorrelation strength parameter ($\gamma \in [0, 1]$). Within the initial time window, the trend follows a linear pattern, either increasing or decreasing. A random noise $\epsilon$ sampled from a normal distribution was added to the trend component. The formulation is as below and the time lag was fixed at $k = 24$ in all synthetic datasets:

$$\mathbf{a}_{t+1} = \gamma \sum_{\tau=t-k}^{t} \omega_\tau \mathbf{a}_\tau + (1-\gamma)\epsilon_a, \qquad (4)$$

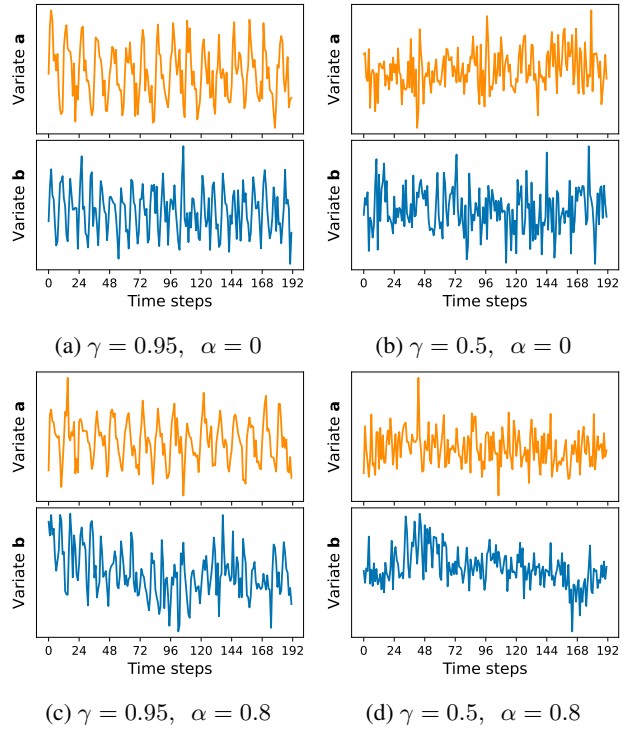

(a) $\gamma = 0.95, \quad \alpha = 0$    (b) $\gamma = 0.5, \quad \alpha = 0$

(c) $\gamma = 0.95, \quad \alpha = 0.8$    (d) $\gamma = 0.5, \quad \alpha = 0.8$

Figure 3: Visualization of synthetic data with different configurations of $\gamma$ and $\alpha$. Figures 3a and 3b show independent variates with high and low autocorrelation, while Figures 3c and 3d depict dependent variates with high and low autocorrelation.

(2) Next, We generated two seasonal components for each variate, defined by specified amplitudes and frequencies. Each seasonal component was represented as a sine wave. The final time series was generated by adding the trend and seasonal components.

For the dependent variates, we initially generated two independent variates using the above process. An additional step was then applied to introduce a specified dependency parameter ($\alpha \in [0, 1]$) between the variates. This parameter controls the strength of the dependency: when $\alpha = 0$, the variates were generated independently, with no dependency between them.

$$\hat{\mathbf{b}}_{t+1} = \alpha \sum_{\tau=t-k}^{t} \hat{\omega}_\tau \mathbf{a}_\tau + (1-\alpha)\mathbf{b}_{t+1}. \qquad (5)$$

In our experiments, we selected $\gamma \in \{0.5, 0.95\}$ to represent low and high levels of autocorrelation, respectively, and $\alpha \in \{0, 0.2, 0.4, 0.8\}$ to capture varying degrees of dependency between variates. Figure 3 illustrates the synthetic signals generated with different levels of $\gamma$ and $\alpha$.

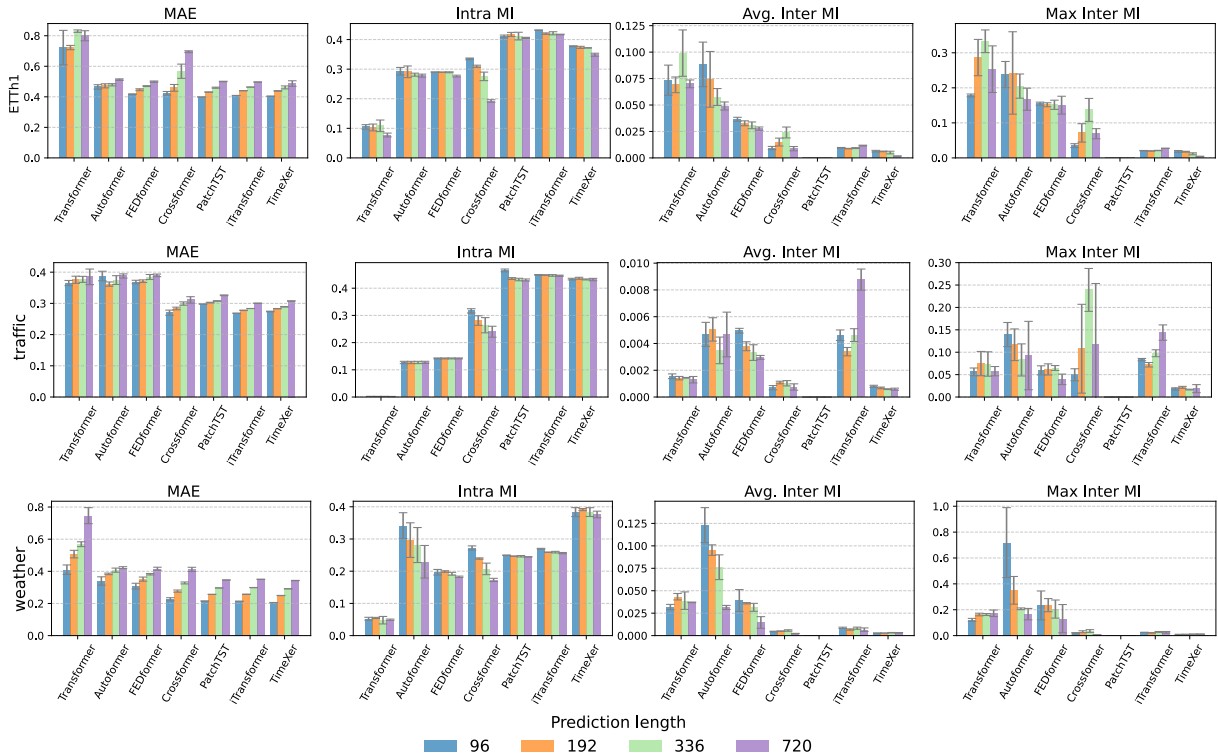

Figure 4: Performance comparison of selected transformer-based models on commonly used benchmark datasets.

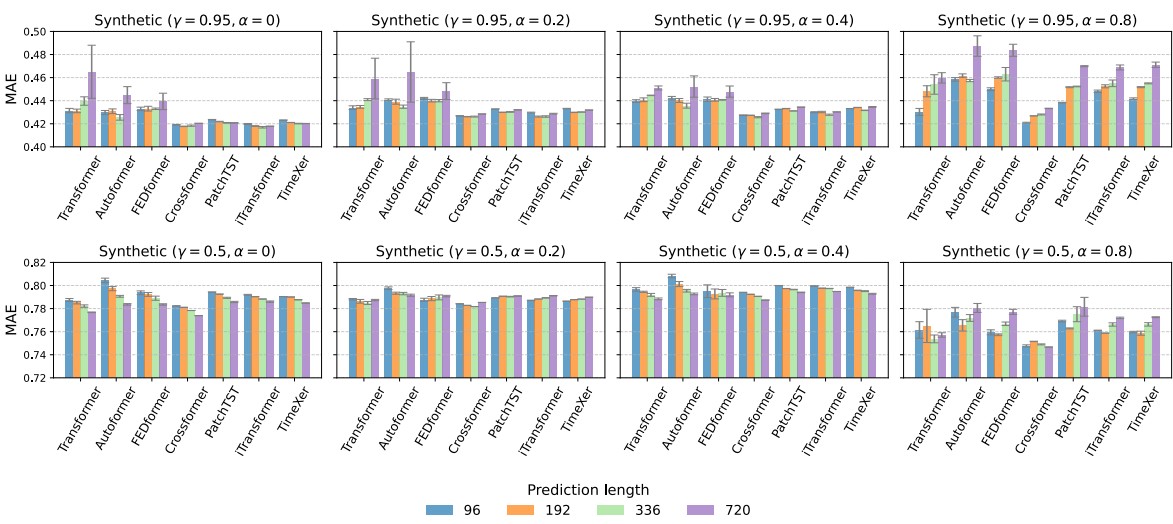

Figure 5: Forecasting performance of selected transformer-based models on synthetic datasets with varying degrees of dependency between and within variates.

## 4.2. Results

In this section, we present and analyze the results of our experiments, organized around the research questions posed in this study.

**1. Why point-wise transformers are generally less competitive in time series forecasting? And why transform-**

**ers with intra-variate attention and those with inter-variate attention often achieve similar performance?**

We begin by comparing the performance of selected transformer-based models on widely used benchmark datasets (Figure 4). The full results of all benchmarks datasets are provided in Figure 1 of the Appendix. These results indicate that transformers employing point-wise to-

Table 2: Correlation between MAE and mutual information scores across various datasets. Additionally, the table includes the average correlation between different variates within each dataset (denoted as Avg Var Corr).

| | ETTh1 | ETTh2 | ETTm1 | ETTm2 | Weather | Traffic | Electricity | Synthetic ($\gamma = 0.95$) | | | | Synthetic ($\gamma = 0.5$) | | | |
|---|---|---|---|---|---|---|---|---|---|---|---|---|---|---|---|
| | | | | | | | | $\alpha = 0$ | $\alpha = 0.2$ | $\alpha = 0.4$ | $\alpha = 0.8$ | $\alpha = 0$ | $\alpha = 0.2$ | $\alpha = 0.4$ | $\alpha = 0.8$ |
| Avg Var Corr | 0.222 | 0.325 | 0.224 | 0.324 | 0.296 | 0.564 | 0.489 | 0.017 | 0.034 | 0.061 | 0.048 | 0.016 | 0.012 | 0.019 | 0.031 |
| MAE - Intra | -0.875 | -0.935 | -0.794 | -0.781 | -0.715 | -0.87 | -0.739 | -0.742 | -0.515 | -0.6 | 0.089 | 0.8 | 0.792 | 0.725 | 0.695 |
| MAE - Avg Inter | 0.616 | 0.725 | 0.477 | 0.656 | 0.376 | 0.266 | 0.596 | 0.681 | 0.728 | 0.788 | 0.292 | 0.457 | 0.457 | 0.54 | 0.12 |
| MAE - Max Inter | 0.64 | 0.784 | 0.544 | 0.802 | 0.32 | 0.11 | 0.347 | 0.681 | 0.728 | 0.788 | 0.292 | 0.457 | 0.457 | 0.54 | 0.12 |

kens are generally less effective at capturing patterns within a single variate, as evidenced by their lower Intra MI scores. In contrast, these models are more sensitive to inter-variate influences, leading to higher Avg Inter MI and Max Inter MI scores. However, despite these higher Inter MI scores, they do not significantly improve the forecasting performance. This suggests that the limited performance of point-wise transformers is primarily due to their inability to effectively capture intra-variate patterns. Additionally, these results indicate that the benchmark datasets are largely self-dependent, with variates showing minor inter-variate dependencies.

To validate this hypothesis, we conducted experiments on synthetic datasets with varying levels of intra- and inter-variate dependencies. The results in Table 2 reveal a **strong negative correlation** between MAE and Intra MI scores on benchmark datasets, as well as on synthetic datasets with high autocorrelation ($\gamma = 0.95$) and low inter-variate dependencies ($\alpha \leq 0.4$). In contrast, on synthetic datasets with low autocorrelation ($\gamma = 0.5$), the results show a positive correlation between MAE and Intra MI scores. MSE follows the same pattern as MAE, so it is omitted here for brevity. Accordingly, transformers that perform exceptionally well on benchmark datasets are less effective on synthetic datasets characterized by high inter-variate dependencies ($\alpha = 0.8$) or low autocorrelation ($\gamma = 0.5$). As illustrated in Figure 5, Crossformer outperforms PatchTST, iTransformer, and TimeXer in these scenarios. **These findings suggest that the performance of transformers in those benchmarks of time series forecasting is heavily influenced by the strength of intra-variate patterns. As a result, point-wise transformers are less competitive due to their limited ability to capture such patterns**.

We also observe that, despite having inter-variate attention mechanisms, iTransformer and TimeXer generally exhibit lower Inter MI scores compared to other models (Figure 4). **This suggests that inter-variate patterns have a limited impact on forecasting benchmarks**, offering insight into the second research question: Why do transformers with inter-variate attention often perform similarly to those with intra-variate attention? The reason is that both primarily focus on capturing intra-variate patterns.

Furthermore, the average correlation between variates within each dataset (Avg Var Corr) is not a reliable indicator of the dependency strength between variates. This is evident in the synthetic datasets, where the correlation between variates remains close to zero and is unaffected by the dependency strength parameter $\alpha$. This also explains why datasets with higher variate correlation do not necessarily exhibit stronger inter-variate dependencies.

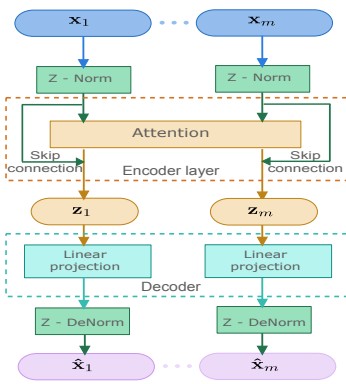

Figure 6: Demonstration of iTransformer architecture.

## 2. Why do transformers with basic attention mechanisms excel in time series forecasting? Which components in the basic transformer's architecture contribute most to the success in time series forecasting?

Given the importance of intra-variate patterns in time series forecasting, we hypothesize that the key components of a transformer's architecture are those that most effectively enhance its ability to capture intra-variate dependencies. To validate this hypothesis, we conducted an ablation study on the iTransformer (Liu et al., 2023) model, which includes the vanilla transformer encoder and a variant-independent linear decoder (Figure 6). PatchTST (Nie et al., 2023) has a similar architecture to iTransformer, but its attention mechanism operates within each variate rather than across all variates, forcing it to focus entirely on capturing intra-variate patterns. In our experiments, we modified the model by removing the skip connections in the encoder layers and/or replacing the variate-independent decoder with a variate-dependent one. Specifically, for the benchmark

Table 3: Ablation study on the iTransformer model architecture. "w/o SC" refers to the model without skip connections in the encoder layers, "VD-De" denotes the use of a variate-dependent decoder, "w/o SC & VD-De" indicates both modifications applied, and "Original" represents the unmodified iTransformer model. The results are averaged over different prediction length, best results are highlighted in red.

| Metric | iTransformer | Weather | ETTh1 | ETTh2 | ETTm1 | ETTm2 | Electricity | Traffic | Synthetic ($\gamma$=0.95) | | | | Synthetic ($\gamma$=0.5) | | | |
|---|---|---|---|---|---|---|---|---|---|---|---|---|---|---|---|---|
| | | | | | | | | | $\alpha = 0$ | $\alpha = 0.2$ | $\alpha = 0.4$ | $\alpha = 0.8$ | $\alpha = 0$ | $\alpha = 0.2$ | $\alpha = 0.4$ | $\alpha = 0.8$ |
| MAE | w/o SC | 0.295 | 0.472 | 0.420 | 0.421 | 0.340 | 0.320 | 0.591 | 0.443 | 0.436 | 0.434 | 0.455 | 0.789 | 0.789 | 0.798 | 0.764 |
| | VD-De | 0.282 | 0.462 | 0.417 | 0.413 | 0.338 | 0.268 | 0.29 | 0.419 | 0.428 | 0.430 | 0.461 | 0.788 | 0.788 | 0.795 | 0.761 |
| | w/o SC & VD-De | 0.287 | 0.502 | 0.423 | 0.424 | 0.342 | 0.363 | 0.631 | 0.419 | 0.428 | 0.430 | 0.449 | 0.788 | 0.789 | 0.799 | 0.763 |
| | Original | 0.280 | 0.452 | 0.407 | 0.412 | 0.335 | 0.266 | 0.283 | 0.418 | 0.428 | 0.430 | 0.456 | 0.789 | 0.789 | 0.797 | 0.765 |
| MSE | w/o SC | 0.275 | 0.493 | 0.397 | 0.426 | 0.297 | 0.229 | 1.017 | 0.315 | 0.298 | 0.297 | 0.326 | 0.969 | 0.978 | 0.995 | 0.918 |
| | VD-De | 0.257 | 0.473 | 0.397 | 0.411 | 0.293 | 0.173 | 0.425 | 0.276 | 0.286 | 0.291 | 0.336 | 0.969 | 0.973 | 0.987 | 0.912 |
| | w/o SC & VD-De | 0.263 | 0.536 | 0.401 | 0.429 | 0.299 | 0.283 | 1.121 | 0.275 | 0.285 | 0.290 | 0.318 | 0.969 | 0.977 | 0.995 | 0.916 |
| | Original | 0.26 | 0.46 | 0.383 | 0.409 | 0.291 | 0.175 | 0.422 | 0.274 | 0.286 | 0.290 | 0.330 | 0.970 | 0.978 | 0.993 | 0.921 |

datasets, we added a 2D convolutional layer before the variate-independent linear projection in the decoder. While for the synthetic datasets, we added a flattened linear layer. These changes were introduced to enable interactions between variates in the decoder.

The results in Table 3, show that removing skip connections notably degrades the model's performance across all benchmark datasets, with particularly significant degradation on Electricity (MAE increases from 0.266 to 0.320) and Traffic (MAE increases from 0.283 to 0.591). However, on the synthetic datasets, this degradation becomes negligible while the self-dependency becomes less (i.e. the autocorrelation $\gamma$ becomes lower, or the inter-variate dependency $\alpha$ becomes higher). This observation suggests that **skip connections are crucial for capturing intra-variate dependencies but may limit the model's ability to capture inter-variate patterns**. On the other hand, replacing the variate-independent decoder with a variate-dependent one has minor impact on performance for benchmark datasets. However, it improves performance on synthetic datasets with high inter-variate dependencies ($\alpha = 0.8$) under low autocorrelation ($\gamma = 0.5$). This indicates that **the variate-dependent decoder enhances the model's capacity to capture inter-variate interactions, especially in scenarios with strong inter-variate dependencies**. A more advanced decoder design could potentially further enhance performance on datasets with strong inter-variate dependencies, which we leave as a direction for future work.

Additionally, we observed that applying Z-score normalization and denormalization to the model's input and output significantly improves performance on benchmark datasets. To evaluate this, we tested four models, Crossformer, PatchTST, iTransformer, and TimeXer, with and without Z-score normalization/denormalization across all datasets. Point-wise models were excluded from this test

because the normalization is incompatible with their architecture, which will be applied along the time dimension instead of the variate dimension. As shown in Table 4, Z-score normalization has the opposite effect on synthetic datasets compared to benchmark datasets. This suggests that the benchmark datasets are largely stationary, making it unreliable to evaluate models solely based on their performance on these datasets. Such comparisons may not accurately reflect their ability to handle non-stationary data. Therefore, the use of such normalization techniques should be assessed on a case-by-case basis, depending on the specific characteristics of the dataset.

## 5. Real-world Applications

Our experiments revealed that the performance of transformers in time series forecasting benchmarks rely heavily on their ability to capture intra-variate patterns. We also found that current benchmark datasets are largely self-dependent and stationary, with minor inter-variate dependencies. As a result, models that excel on these datasets are less effective on synthetic datasets with strong inter-variate dependencies or low autocorrelation.

In real-world applications, time series data often exhibit more complicated patterns with varying degrees of inter-variate dependencies and autocorrelation. To validate our findings, we examined two real-world healthcare datasets, MINDER and TIHM (Palermo et al., 2023), which contain location-activity data from PLwD (People Living with Dementia). Understanding and forecasting behavioral pattern changes in dementia patients is crucial. However, these datasets differ significantly from both benchmarking and synthetic datasets, creating challenges for state-of-the-art time series forecasting models.

Table 5 shows the correlation between the forecasting metric MAE and mutual information scores on the MINDER

Table 4: The effect of Z-normalization on the performance of various models across benchmark and synthetic datasets. Results are averaged over different prediction lengths, with the best results highlighted in red. The blue bold font in the "Z-Norm" column indicates the original model implementation.

| Metric | Model | Z-Norm | Weather | ETTh1 | ETTh2 | ETTm1 | ETTm2 | Electricity | Traffic | Synthetic ($\gamma = 0.95$) | | | | Synthetic ($\gamma = 0.5$) | | | |
|---|---|---|---|---|---|---|---|---|---|---|---|---|---|---|---|---|---|
| | | | | | | | | | | $\alpha = 0$ | $\alpha = 0.2$ | $\alpha = 0.4$ | $\alpha = 0.8$ | $\alpha = 0$ | $\alpha = 0.2$ | $\alpha = 0.4$ | $\alpha = 0.8$ |
| MAE | Crossformer | w/ | 0.271 | 0.447 | 0.424 | 0.401 | 0.330 | 0.264 | 0.288 | 0.424 | 0.433 | 0.434 | 0.452 | 0.787 | 0.788 | 0.797 | 0.765 |
| | | **w/o** | 0.311 | 0.536 | 0.997 | 0.546 | 0.803 | 0.283 | 0.291 | 0.419 | 0.427 | 0.427 | 0.427 | 0.779 | 0.784 | 0.791 | 0.749 |
| | PatchTST | **w/** | 0.279 | 0.447 | 0.409 | 0.402 | 0.331 | 0.297 | 0.309 | 0.422 | 0.431 | 0.433 | 0.453 | 0.790 | 0.790 | 0.797 | 0.772 |
| | | w/o | 0.3 | 0.483 | 0.520 | 0.434 | 0.405 | 0.299 | 0.316 | 0.417 | 0.424 | 0.425 | 0.428 | 0.783 | 0.785 | 0.791 | 0.756 |
| | iTransformer | **w/** | 0.28 | 0.452 | 0.407 | 0.412 | 0.335 | 0.266 | 0.282 | 0.418 | 0.428 | 0.430 | 0.456 | 0.789 | 0.789 | 0.797 | 0.765 |
| | | w/o | 0.299 | 0.495 | 0.668 | 0.457 | 0.573 | 0.282 | 0.315 | 0.414 | 0.423 | 0.423 | 0.429 | 0.781 | 0.784 | 0.792 | 0.751 |
| | TimeXer | **w/** | 0.272 | 0.448 | 0.404 | 0.398 | 0.323 | 0.269 | 0.288 | 0.421 | 0.431 | 0.433 | 0.455 | 0.788 | 0.788 | 0.796 | 0.764 |
| | | w/o | 0.311 | 0.493 | 0.891 | 0.473 | 0.715 | 0.286 | 0.326 | 0.416 | 0.425 | 0.426 | 0.429 | 0.780 | 0.783 | 0.789 | 0.750 |
| MSE | Crossformer | w/ | 0.241 | 0.456 | 0.407 | 0.394 | 0.286 | 0.171 | 0.471 | 0.281 | 0.292 | 0.296 | 0.321 | 0.966 | 0.976 | 0.992 | 0.922 |
| | | **w/o** | 0.256 | 0.572 | 1.742 | 0.571 | 1.715 | 0.189 | 0.566 | 0.275 | 0.285 | 0.288 | 0.288 | 0.946 | 0.961 | 0.977 | 0.883 |
| | PatchTST | **w/** | 0.256 | 0.453 | 0.386 | 0.388 | 0.286 | 0.208 | 0.482 | 0.280 | 0.291 | 0.294 | 0.322 | 0.973 | 0.981 | 0.992 | 0.940 |
| | | w/o | 0.246 | 0.489 | 0.600 | 0.423 | 0.379 | 0.204 | 0.595 | 0.273 | 0.281 | 0.284 | 0.289 | 0.953 | 0.967 | 0.978 | 0.899 |
| | iTransformer | **w/** | 0.260 | 0.460 | 0.383 | 0.409 | 0.291 | 0.175 | 0.422 | 0.274 | 0.286 | 0.290 | 0.330 | 0.970 | 0.978 | 0.993 | 0.921 |
| | | w/o | 0.250 | 0.508 | 0.842 | 0.450 | 0.653 | 0.182 | 0.571 | 0.268 | 0.279 | 0.282 | 0.292 | 0.950 | 0.964 | 0.978 | 0.888 |
| | TimeXer | **w/** | 0.242 | 0.455 | 0.379 | 0.384 | 0.276 | 0.170 | 0.470 | 0.279 | 0.291 | 0.295 | 0.327 | 0.968 | 0.975 | 0.989 | 0.921 |
| | | w/o | 0.257 | 0.496 | 1.555 | 0.467 | 1.156 | 0.187 | 0.604 | 0.272 | 0.282 | 0.285 | 0.291 | 0.948 | 0.961 | 0.972 | 0.885 |

Table 5: Correlation between MAE and mutual information scores on real-world health care datasets.

| | Avg Var Corr | MAE - Intra | MAE - Avg Inter | MAE - Max Inter |
|---|---|---|---|---|
| MINDER | 0.185 | -0.358 | 0.882 | 0.873 |
| TIHM | 0.321 | -0.703 | -0.11 | -0.05 |

Table 6: Forecasting performance of iTransformer variants on TIHM and MINDER datasets.

| | MAE | | MSE | |
|---|---|---|---|---|
| iTransformer | Original | VD-De | Original | VD-De |
| TIHM | 0.623 | 0.610 | 0.835 | 0.807 |
| MINDER | 0.419 | 0.431 | 0.721 | 0.726 |

and TIHM datasets across all selected models. Despite the two datasets having similar variates, their properties differ significantly, likely because the data was collected from different participants. Based on these correlations, we hypothesized that models with a stronger capacity for capturing inter-variate dependencies would perform better on the TIHM dataset. The results in Table 6 support this hypothesis, showing that incorporating a variate-dependent decoder enhances the iTransformer model's performance on the THIM dataset but negatively impacts its performance on the MINDER dataset. These findings suggest that a single, universal model architecture may not be suitable for all datasets or applications, as different datasets exhibit distinct dependency structures and require tailored modeling approaches.

Moreover, the results in Table 7 show that Z-normalization has opposing effects on MAE and MSE for the TIHM dataset, with MSE deteriorating while MAE improving. This effect likely arises because normalization alters the scale and distribution of the input data, which may disrupt the model's ability to capture larger errors while making it more effective in reducing absolute errors. These findings highlight the importance of carefully selecting evaluation metrics based on application requirements, as different metrics may prioritize different aspects of forecasting performance, leading to varying interpretations of model effectiveness.

Table 7: Z-normalization shows opposing effects on MAE and MSE for TIHM dataset.

| Model | | Crossformer | | PatchTST | | iTransformer | | TimeXer | |
|---|---|---|---|---|---|---|---|---|---|
| Z-Norm | | w/ | w/o | w/ | w/o | w/ | w/o | w/ | w/o |
| TIHM | MAE | 0.739 | 0.748 | 0.612 | 0.614 | 0.622 | 0.634 | 0.61 | 0.614 |
| | MSE | 1.038 | 1.011 | 0.829 | 0.82 | 0.835 | 0.834 | 0.817 | 0.8 |

Future research should explore adaptive evaluation techniques and dynamic modeling of intra- and inter-variate dependencies. Expanding benchmarks with diverse, nonstationary datasets will further enhance model generalizability in real-world applications.

## Acknowledgements

This research is supported by the UK Dementia Research Institute [award number UK DRI-7002] through UK DRI Ltd, principally funded by the Medical Research Council (MRC), and the UKRI Engineering and Physical Sciences Research Council (EPSRC) PROTECT Project (grant number: EP/W031892/1). Infrastructure support for this research was provided by the NIHR Imperial Biomedical Research Centre (BRC) and the UKRI Medical Research Council (MRC). P.B. is also funded by the Great Ormond Street Hospital and the Royal Academy of Engineering (grant number: RCSRF2324-18-69). We would also like to thank our reviewers for their valuable comments and suggestions, which helped improve the quality and clarity of this manuscript.

## Impact Statement

This paper presents work whose goal is to advance the field of Machine Learning. There are several potential societal benefits of our work and potential risks which are mitigated in our clinical study, none which we feel must be specifically highlighted here.

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

# A. Appendix

## A.1. Details of the datasets

We provide key attributes of all datasets employed in this study in Table 1. The datasets include seven benchmark datasets (ETTh1, ETTh2, ETTm1, ETTm2, Weather, Traffic, and Electricity), eight synthetic dataset, and two real-world healthcare datasets (TIHM and MINDER). The benchmark datasets are publicly available on GitHub (`https://github.com/thuml/Time-Series-Library/tree/main`). The healthcare dataset TIHM is available on Zenodo (`https://doi.org/10.5281/zenodo.7622128`). The MINDER dataset is not publicly available due to privacy protocols and has a format and variates similar to those of the TIHM dataset.

### A.1.1. REAL-WORLD HEALTHCARE DATASETS

**TIMH dataset** The TIHM dataset consists of five interconnected tables, Activity, Sleep, Physiology, Labels, and Demographics, capturing various aspects of remote healthcare monitoring. Each table includes timestamps and participant UUIDs for cross-referencing and synchronization. The dataset includes 56 participants, all over 50 years old with a verified diagnosis of dementia or mild cognitive impairment, who provided informed consent. Each participant had a caregiver or study partner, and individuals with severe psychiatric conditions or terminal illnesses were excluded. The study recorded an average of 50 days of data per participant.

For this study, we selected the participant with the longest recording, spanning three months from April 1, 2019, to June 30, 2019. We focused on the Activity table, which tracks in-home movement using motion and door sensors. Specifically, the dataset used in this study includes activity records from eight locations: back door, bathroom, bedroom, fridge door, front door, hallway, kitchen, and lounge. The data was aggregated on an hourly basis by summing the activity counts for each location.

**MINDER dataset** The MINDER dataset, an ongoing project, collects in-home data from 117 participants over the age of 50, all with a clinical diagnosis of dementia or mild cognitive impairment (MCI). Each participant had received either past or ongoing psychiatric treatment. For this study, we selected the participant with the longest recording, spanning two years from December 1, 2022, to November 30, 2024. The MINDER dataset is similar in format to TIHM, and we also focused exclusively on the Activity table. Specifically, the dataset for this work includes activity records from eight locations: WC, bathroom, bedroom, hallway, front door, kitchen, and lounge.

## A.2. Full results of all selected models on benchmark datasets and synthetic datasets

Figure 1 shows evaluations of all selected models on all benchmark datasets using MAE, Intra MI, Avg Inter MI, and Max Inter MI scores.

Table 2 shows evaluations of variants of iTransformer model on all benchmark datasets using MAE, Intra MI, Avg Inter MI, and Max Inter MI scores. Table 3 gives the same evaluation on synthetic datasets. Tables 4 to 7 provide MAE and MSE of selected models on benchmark and synthetic datasets for the Z-normalization ablation study.

Table 1: Key attributes of the datasets: Dim represents the number of variates, Dataset Size indicates the total time points across (Train, Validation, Test) splits, Prediction Length specifies the number of future time points to predict (with four settings per dataset), and Frequency denotes the sampling interval of the time points.

| Dataset | Dim | Prediction Length | Dataset Size | Frequency | Information |
|---|---|---|---|---|---|
| ETTh1, ETTh2 | 7 | {96, 192, 336, 720} | (8545, 2881, 2881) | Hourly | Electricity |
| ETTm1, ETTm2 | 7 | {96, 192, 336, 720} | (34465, 11521, 11521) | 15 min | Electricity |
| Weather | 21 | {96, 192, 336, 720} | (36792, 5271, 10540) | 10 min | Weather |
| Traffic | 862 | {96, 192, 336, 720} | (12185, 1757, 3509) | Hourly | Transportation |
| ECL | 321 | {96, 192, 336, 720} | (18317, 2633, 5261) | Hourly | Electricity |
| Synthetic (8 subsets) | 2 | {96, 192, 336, 720} | (6132, 876, 1752) | Hourly | Synthetic |
| TIHM | 8 | {48, 96, 192} | (1496, 427, 213) | Hourly | Healthcare |
| MINDER | 8 | {48, 96, 192} | (12281, 3508, 1754) | Hourly | Healthcare |

Table 2: Results of variants of iTranformer model on benchmark datasets. "w/o SC" refers to the model without skip connections in the encoder layers, "VD-De" denotes the use of a variate-dependent decoder, "w/o SC & VD-De" indicates both modifications applied, and "Original" represents the unmodified iTransformer model.

| Dataset | Pred length | MAE | | | | MSE | | | |
|---|---|---|---|---|---|---|---|---|---|
| | | w/o SC | VD-De | w/o SC & VD-De | Original | w/o SC | VD-De | w/o SC & VD-De | Original |
| Weather | 96 | 0.235 ± 0.001 | 0.218 ± 0.0 | 0.226 ± 0.002 | 0.214 ± 0.001 | 0.193 ± 0.001 | 0.174 ± 0.0 | 0.181 ± 0.001 | 0.175 ± 0.001 |
| | 192 | 0.277 ± 0.001 | 0.259 ± 0.001 | 0.265 ± 0.003 | 0.257 ± 0.001 | 0.245 ± 0.001 | 0.222 ± 0.001 | 0.229 ± 0.003 | 0.224 ± 0.001 |
| | 336 | 0.311 ± 0.001 | 0.3 ± 0.001 | 0.303 ± 0.0 | 0.299 ± 0.001 | 0.294 ± 0.001 | 0.278 ± 0.001 | 0.283 ± 0.001 | 0.281 ± 0.001 |
| | 720 | 0.357 ± 0.001 | 0.351 ± 0.002 | 0.352 ± 0.001 | 0.35 ± 0.0 | 0.369 ± 0.001 | 0.355 ± 0.001 | 0.358 ± 0.002 | 0.36 ± 0.001 |
| ETTh1 | 96 | 0.433 ± 0.0 | 0.427 ± 0.004 | 0.467 ± 0.015 | 0.409 ± 0.0 | 0.429 ± 0.001 | 0.412 ± 0.006 | 0.478 ± 0.025 | 0.394 ± 0.001 |
| | 192 | 0.461 ± 0.001 | 0.453 ± 0.002 | 0.493 ± 0.014 | 0.44 ± 0.001 | 0.481 ± 0.002 | 0.461 ± 0.003 | 0.528 ± 0.027 | 0.447 ± 0.001 |
| | 336 | 0.482 ± 0.001 | 0.471 ± 0.003 | 0.513 ± 0.009 | 0.464 ± 0.001 | 0.524 ± 0.001 | 0.5 ± 0.003 | 0.565 ± 0.014 | 0.49 ± 0.001 |
| | 720 | 0.51 ± 0.002 | 0.499 ± 0.002 | 0.536 ± 0.002 | 0.497 ± 0.001 | 0.539 ± 0.005 | 0.52 ± 0.004 | 0.575 ± 0.005 | 0.511 ± 0.002 |
| ETTh2 | 96 | 0.366 ± 0.001 | 0.361 ± 0.003 | 0.371 ± 0.002 | 0.35 ± 0.001 | 0.317 ± 0.001 | 0.312 ± 0.005 | 0.321 ± 0.003 | 0.3 ± 0.001 |
| | 192 | 0.412 ± 0.001 | 0.410 ± 0.002 | 0.416 ± 0.002 | 0.399 ± 0.0 | 0.397 ± 0.001 | 0.396 ± 0.003 | 0.401 ± 0.003 | 0.379 ± 0.0 |
| | 336 | 0.445 ± 0.002 | 0.444 ± 0.003 | 0.447 ± 0.002 | 0.433 ± 0.001 | 0.437 ± 0.003 | 0.441 ± 0.006 | 0.44 ± 0.004 | 0.423 ± 0.002 |
| | 720 | 0.455 ± 0.001 | 0.452 ± 0.002 | 0.457 ± 0.002 | 0.447 ± 0.001 | 0.438 ± 0.001 | 0.437 ± 0.004 | 0.441 ± 0.004 | 0.43 ± 0.003 |
| ETTm1 | 96 | 0.392 ± 0.006 | 0.379 ± 0.001 | 0.391 ± 0.002 | 0.378 ± 0.001 | 0.37 ± 0.008 | 0.345 ± 0.001 | 0.365 ± 0.004 | 0.345 ± 0.003 |
| | 192 | 0.406 ± 0.009 | 0.397 ± 0.001 | 0.409 ± 0.003 | 0.395 ± 0.0 | 0.402 ± 0.012 | 0.384 ± 0.001 | 0.402 ± 0.005 | 0.383 ± 0.001 |
| | 336 | 0.426 ± 0.004 | 0.42 ± 0.0 | 0.43 ± 0.001 | 0.418 ± 0.0 | 0.434 ± 0.005 | 0.424 ± 0.0 | 0.439 ± 0.004 | 0.418 ± 0.002 |
| | 720 | 0.46 ± 0.005 | 0.457 ± 0.001 | 0.467 ± 0.003 | 0.457 ± 0.002 | 0.497 ± 0.007 | 0.492 ± 0.004 | 0.509 ± 0.006 | 0.49 ± 0.003 |
| ETTm2 | 96 | 0.279 ± 0.002 | 0.275 ± 0.0 | 0.282 ± 0.001 | 0.271 ± 0.0 | 0.193 ± 0.001 | 0.188 ± 0.001 | 0.196 ± 0.002 | 0.185 ± 0.001 |
| | 192 | 0.317 ± 0.002 | 0.315 ± 0.001 | 0.318 ± 0.001 | 0.311 ± 0.001 | 0.256 ± 0.001 | 0.253 ± 0.002 | 0.258 ± 0.001 | 0.25 ± 0.001 |
| | 336 | 0.356 ± 0.003 | 0.353 ± 0.002 | 0.356 ± 0.001 | 0.352 ± 0.001 | 0.32 ± 0.005 | 0.316 ± 0.002 | 0.321 ± 0.002 | 0.316 ± 0.001 |
| | 720 | 0.410 ± 0.001 | 0.408 ± 0.002 | 0.41 ± 0.002 | 0.406 ± 0.001 | 0.418 ± 0.001 | 0.415 ± 0.003 | 0.42 ± 0.002 | 0.412 ± 0.002 |
| Electricity | 96 | 0.287 ± 0.004 | 0.240 ± 0.0 | 0.35 ± 0.0 | 0.24 ± 0.0 | 0.189 ± 0.005 | 0.144 ± 0.0 | 0.269 ± 0.0 | 0.148 ± 0.0 |
| | 192 | 0.306 ± 0.004 | 0.259 ± 0.0 | 0.454 ± 0.001 | 0.256 ± 0.0 | 0.211 ± 0.005 | 0.165 ± 0.0 | 0.271 ± 0.001 | 0.165 ± 0.0 |
| | 336 | 0.325 ± 0.003 | 0.274 ± 0.003 | 0.362 ± 0.001 | 0.27 ± 0.001 | 0.232 ± 0.004 | 0.177 ± 0.002 | 0.280 ± 0.001 | 0.178 ± 0.001 |
| | 720 | 0.360 ± 0.006 | 0.3 ± 0.001 | 0.384 ± 0.002 | 0.299 ± 0.001 | 0.283 ± 0.01 | 0.206 ± 0.001 | 0.311 ± 0.002 | 0.21 ± 0.001 |
| Traffic | 96 | 0.663 ± 0.005 | 0.276 ± 0.001 | 0.799 ± 0.005 | 0.268 ± 0.0 | 1.123 ± 0.016 | 0.396 ± 0.001 | 1.406 ± 0.002 | 0.392 ± 0.001 |
| | 192 | 0.559 ± 0.004 | 0.284 ± 0.001 | 0.567 ± 0.006 | 0.278 ± 0.001 | 0.945 ± 0.008 | 0.415 ± 0.002 | 1.016 ± 0.004 | 0.413 ± 0.002 |
| | 336 | 0.572 ± 0.006 | 0.291 ± 0.002 | 0.581 ± 0.003 | 0.284 ± 0.0 | 1.004 ± 0.01 | 0.427 ± 0.001 | 1.029 ± 0.001 | 0.426 ± 0.0 |
| | 720 | 0.569 ± 0.008 | 0.307 ± 0.001 | 0.571 ± 0.004 | 0.3 ± 0.001 | 0.996 ± 0.007 | 0.461 ± 0.001 | 1.03 ± 0.001 | 0.458 ± 0.001 |

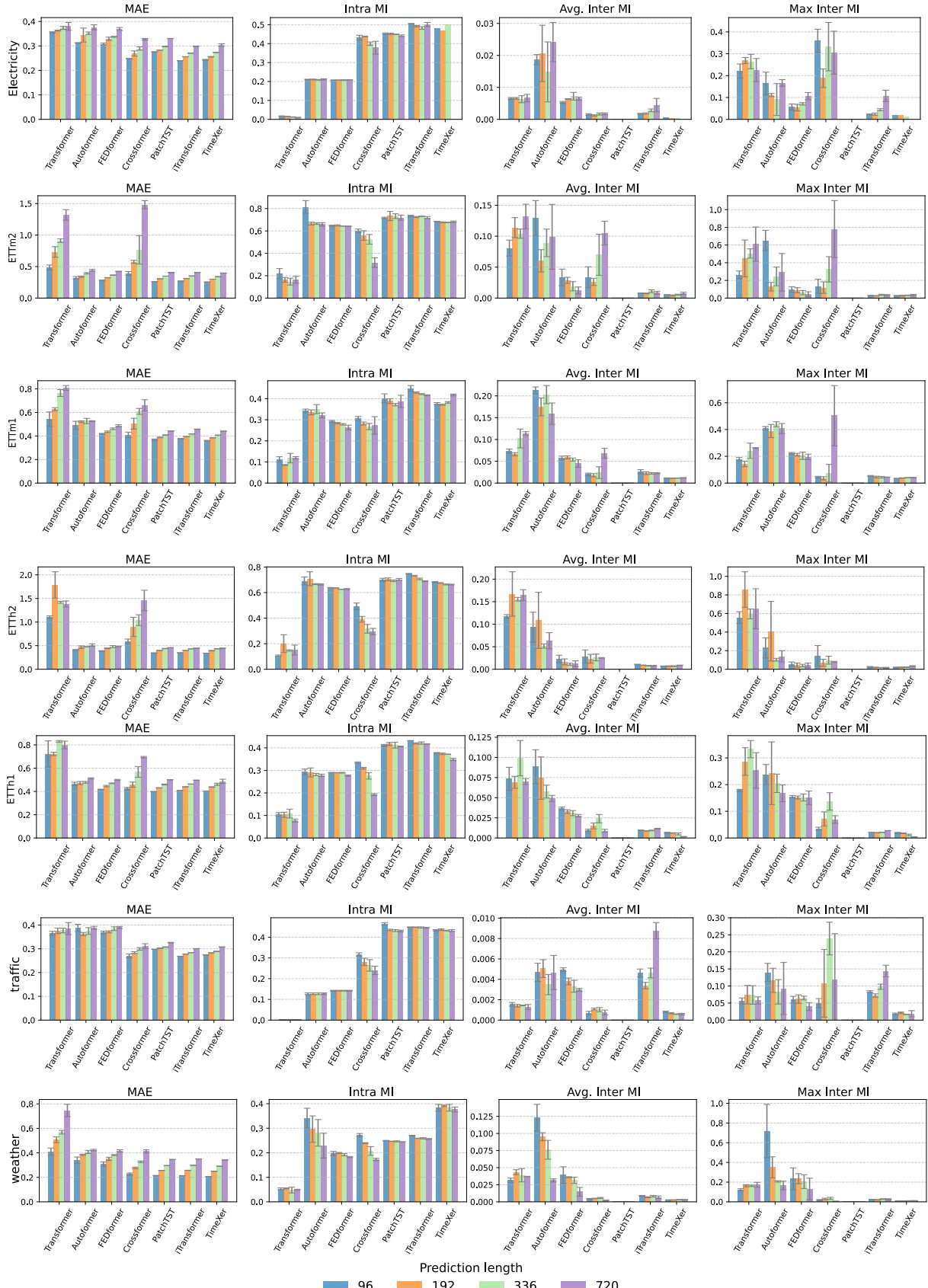

Figure 1: Performance comparison of selected transformer-based models on commonly used benchmark datasets.

Table 3: Results of variants of iTranformer model on synthetic datasets. "w/o SC" refers to the model without skip connections in the encoder layers, "VD-De" denotes the use of a variate-dependent decoder, "w/o SC & VD-De" indicates both modifications applied, and "Original" represents the unmodified iTransformer model.

| Dataset | Pred length | MAE | | | | MSE | | | |
|---|---|---|---|---|---|---|---|---|---|
| | | w/o SC | VD-De | w/o SC & VD-De | Original | w/o SC | VD-De | w/o SC & VD-De | Original |
| Synthetic $\gamma = 0.95$ $\alpha = 0$ | 96 | $0.509 \pm 0.071$ | $0.422 \pm 0.0$ | $0.42 \pm 0.0$ | $0.42 \pm 0.0$ | $0.426 \pm 0.12$ | $0.28 \pm 0.0$ | $0.278 \pm 0.0$ | $0.277 \pm 0.0$ |
| | 192 | $0.421 \pm 0.001$ | $0.419 \pm 0.0$ | $0.419 \pm 0.0$ | $0.418 \pm 0.0$ | $0.279 \pm 0.001$ | $0.277 \pm 0.0$ | $0.275 \pm 0.0$ | $0.275 \pm 0.0$ |
| | 336 | $0.420 \pm 0.0$ | $0.418 \pm 0.0$ | $0.418 \pm 0.0$ | $0.417 \pm 0.0$ | $0.277 \pm 0.0$ | $0.274 \pm 0.0$ | $0.274 \pm 0.0$ | $0.273 \pm 0.001$ |
| | 720 | $0.421 \pm 0.0$ | $0.418 \pm 0.0$ | $0.419 \pm 0.0$ | $0.418 \pm 0.0$ | $0.277 \pm 0.0$ | $0.273 \pm 0.0$ | $0.273 \pm 0.0$ | $0.272 \pm 0.001$ |
| Synthetic $\gamma = 0.95$ $\alpha = 0.2$ | 96 | $0.452 \pm 0.002$ | $0.43 \pm 0.004$ | $0.429 \pm 0.0$ | $0.43 \pm 0.0$ | $0.322 \pm 0.004$ | $0.289 \pm 0.0$ | $0.287 \pm 0.0$ | $0.288 \pm 0.0$ |
| | 192 | $0.431 \pm 0.0$ | $0.426 \pm 0.0$ | $0.426 \pm 0.0$ | $0.426 \pm 0.0$ | $0.289 \pm 0.0$ | $0.283 \pm 0.0$ | $0.283 \pm 0.0$ | $0.283 \pm 0.001$ |
| | 336 | $0.43 \pm 0.0$ | $0.426 \pm 0.0$ | $0.426 \pm 0.0$ | $0.426 \pm 0.001$ | $0.29 \pm 0.0$ | $0.284 \pm 0.0$ | $0.284 \pm 0.0$ | $0.284 \pm 0.001$ |
| | 720 | $0.432 \pm 0.0$ | $0.428 \pm 0.0$ | $0.429 \pm 0.0$ | $0.429 \pm 0.0$ | $0.292 \pm 0.0$ | $0.287 \pm 0.0$ | $0.287 \pm 0.0$ | $0.287 \pm 0.0$ |
| Synthetic $\gamma = 0.95$ $\alpha = 0.4$ | 96 | $0.436 \pm 0.001$ | $0.432 \pm 0.001$ | $0.431 \pm 0.0$ | $0.43 \pm 0.0$ | $0.299 \pm 0.001$ | $0.292 \pm 0.001$ | $0.29 \pm 0.0$ | $0.29 \pm 0.001$ |
| | 192 | $0.435 \pm 0.0$ | $0.432 \pm 0.0$ | $0.429 \pm 0.0$ | $0.43 \pm 0.001$ | $0.296 \pm 0.0$ | $0.292 \pm 0.0$ | $0.289 \pm 0.0$ | $0.29 \pm 0.001$ |
| | 336 | $0.432 \pm 0.0$ | $0.427 \pm 0.0$ | $0.428 \pm 0.0$ | $0.428 \pm 0.001$ | $0.295 \pm 0.0$ | $0.287 \pm 0.0$ | $0.289 \pm 0.0$ | $0.289 \pm 0.001$ |
| | 720 | $0.435 \pm 0.001$ | $0.429 \pm 0.0$ | $0.432 \pm 0.0$ | $0.43 \pm 0.0$ | $0.299 \pm 0.001$ | $0.29 \pm 0.0$ | $0.293 \pm 0.0$ | $0.292 \pm 0.0$ |
| Synthetic $\gamma = 0.95$ $\alpha = 0.8$ | 96 | $0.444 \pm 0.001$ | $0.447 \pm 0.002$ | $0.435 \pm 0.0$ | $0.448 \pm 0.001$ | $0.308 \pm 0.001$ | $0.312 \pm 0.003$ | $0.297 \pm 0.0$ | $0.317 \pm 0.002$ |
| | 192 | $0.451 \pm 0.0$ | $0.456 \pm 0.002$ | $0.445 \pm 0.0$ | $0.452 \pm 0.001$ | $0.322 \pm 0.001$ | $0.33 \pm 0.003$ | $0.314 \pm 0.0$ | $0.325 \pm 0.002$ |
| | 336 | $0.454 \pm 0.001$ | $0.462 \pm 0.005$ | $0.45 \pm 0.001$ | $0.455 \pm 0.003$ | $0.326 \pm 0.001$ | $0.337 \pm 0.008$ | $0.319 \pm 0.001$ | $0.328 \pm 0.004$ |
| | 720 | $0.469 \pm 0.001$ | $0.48 \pm 0.002$ | $0.466 \pm 0.001$ | $0.469 \pm 0.002$ | $0.348 \pm 0.001$ | $0.366 \pm 0.004$ | $0.343 \pm 0.001$ | $0.348 \pm 0.003$ |
| Synthetic $\gamma = 0.5$ $\alpha = 0$ | 96 | $0.791 \pm 0.0$ | $0.789 \pm 0.001$ | $0.79 \pm 0.0$ | $0.792 \pm 0.0$ | $0.975 \pm 0.0$ | $0.972 \pm 0.001$ | $0.973 \pm 0.0$ | $0.977 \pm 0.0$ |
| | 192 | $0.79 \pm 0.001$ | $0.79 \pm 0.0$ | $0.79 \pm 0.0$ | $0.79 \pm 0.0$ | $0.973 \pm 0.0$ | $0.974 \pm 0.001$ | $0.973 \pm 0.0$ | $0.973 \pm 0.0$ |
| | 336 | $0.788 \pm 0.001$ | $0.788 \pm 0.001$ | $0.788 \pm 0.0$ | $0.788 \pm 0.0$ | $0.968 \pm 0.0$ | $0.968 \pm 0.001$ | $0.968 \pm 0.0$ | $0.967 \pm 0.0$ |
| | 720 | $0.786 \pm 0.0$ | $0.784 \pm 0.001$ | $0.785 \pm 0.0$ | $0.786 \pm 0.001$ | $0.963 \pm 0.001$ | $0.96 \pm 0.001$ | $0.961 \pm 0.0$ | $0.964 \pm 0.001$ |
| Synthetic $\gamma = 0.5$ $\alpha = 0.2$ | 96 | $0.786 \pm 0.0$ | $0.786 \pm 0.0$ | $0.787 \pm 0.0$ | $0.787 \pm 0.0$ | $0.975 \pm 0.0$ | $0.972 \pm 0.001$ | $0.975 \pm 0.0$ | $0.976 \pm 0.0$ |
| | 192 | $0.788 \pm 0.0$ | $0.787 \pm 0.0$ | $0.788 \pm 0.0$ | $0.788 \pm 0.0$ | $0.977 \pm 0.0$ | $0.971 \pm 0.0$ | $0.977 \pm 0.0$ | $0.977 \pm 0.0$ |
| | 336 | $0.789 \pm 0.0$ | $0.788 \pm 0.0$ | $0.789 \pm 0.0$ | $0.789 \pm 0.001$ | $0.977 \pm 0.0$ | $0.972 \pm 0.001$ | $0.977 \pm 0.0$ | $0.977 \pm 0.001$ |
| | 720 | $0.791 \pm 0.001$ | $0.791 \pm 0.001$ | $0.791 \pm 0.0$ | $0.791 \pm 0.0$ | $0.981 \pm 0.002$ | $0.979 \pm 0.002$ | $0.981 \pm 0.0$ | $0.981 \pm 0.0$ |
| Synthetic $\gamma = 0.5$ $\alpha = 0.4$ | 96 | $0.802 \pm 0.0$ | $0.798 \pm 0.001$ | $0.802 \pm 0.0$ | $0.8 \pm 0.0$ | $1.003 \pm 0.001$ | $0.994 \pm 0.003$ | $1.004 \pm 0.0$ | $0.999 \pm 0.001$ |
| | 192 | $0.799 \pm 0.0$ | $0.794 \pm 0.0$ | $0.799 \pm 0.0$ | $0.798 \pm 0.0$ | $0.999 \pm 0.001$ | $0.986 \pm 0.001$ | $0.998 \pm 0.0$ | $0.996 \pm 0.0$ |
| | 336 | $0.798 \pm 0.0$ | $0.795 \pm 0.001$ | $0.798 \pm 0.0$ | $0.797 \pm 0.0$ | $0.994 \pm 0.0$ | $0.986 \pm 0.002$ | $0.994 \pm 0.0$ | $0.993 \pm 0.0$ |
| | 720 | $0.795 \pm 0.001$ | $0.794 \pm 0.001$ | $0.795 \pm 0.0$ | $0.795 \pm 0.0$ | $0.986 \pm 0.001$ | $0.984 \pm 0.001$ | $0.985 \pm 0.0$ | $0.985 \pm 0.0$ |
| Synthetic $\gamma = 0.5$ $\alpha = 0.8$ | 96 | $0.761 \pm 0.0$ | $0.757 \pm 0.001$ | $0.76 \pm 0.001$ | $0.761 \pm 0.0$ | $0.91 \pm 0.001$ | $0.9 \pm 0.001$ | $0.908 \pm 0.003$ | $0.911 \pm 0.001$ |
| | 192 | $0.758 \pm 0.0$ | $0.755 \pm 0.001$ | $0.757 \pm 0.001$ | $0.759 \pm 0.001$ | $0.905 \pm 0.001$ | $0.898 \pm 0.001$ | $0.904 \pm 0.002$ | $0.907 \pm 0.0$ |
| | 336 | $0.765 \pm 0.001$ | $0.762 \pm 0.001$ | $0.763 \pm 0.001$ | $0.766 \pm 0.001$ | $0.921 \pm 0.003$ | $0.915 \pm 0.002$ | $0.918 \pm 0.002$ | $0.925 \pm 0.004$ |
| | 720 | $0.771 \pm 0.001$ | $0.771 \pm 0.003$ | $0.771 \pm 0.001$ | $0.772 \pm 0.001$ | $0.936 \pm 0.003$ | $0.935 \pm 0.007$ | $0.936 \pm 0.002$ | $0.939 \pm 0.002$ |

Table 4: The effect of Z-normalization on MAE of various models across benchmark datasets.

| Metric | Model | Z-Norm | Pred length | Weather | ETTh1 | ETTh2 | ETTm1 | ETTm2 | Electricity | Traffic |
|---|---|---|---|---|---|---|---|---|---|---|
| MAE | Crossformer | w/ | 96 | $0.201 \pm 0.002$ | $0.403 \pm 0.002$ | $0.361 \pm 0.003$ | $0.357 \pm 0.001$ | $0.256 \pm 0.0$ | $0.232 \pm 0.002$ | $0.262 \pm 0.006$ |
| | | | 192 | $0.247 \pm 0.001$ | $0.432 \pm 0.003$ | $0.409 \pm 0.004$ | $0.385 \pm 0.001$ | $0.308 \pm 0.003$ | $0.251 \pm 0.0$ | $0.273 \pm 0.003$ |
| | | | 336 | $0.292 \pm 0.001$ | $0.45 \pm 0.003$ | $0.459 \pm 0.011$ | $0.410 \pm 0.003$ | $0.349 \pm 0.003$ | $0.269 \pm 0.002$ | $0.294 \pm 0.003$ |
| | | | 720 | $0.345 \pm 0.002$ | $0.502 \pm 0.007$ | $0.467 \pm 0.002$ | $0.452 \pm 0.001$ | $0.408 \pm 0.001$ | $0.303 \pm 0.005$ | $0.321 \pm 0.009$ |
| | | w/o | 96 | $0.226 \pm 0.01$ | $0.423 \pm 0.012$ | $0.592 \pm 0.048$ | $0.409 \pm 0.024$ | $0.392 \pm 0.03$ | $0.248 \pm 0.001$ | $0.27 \pm 0.008$ |
| | | | 192 | $0.277 \pm 0.007$ | $0.459 \pm 0.022$ | $0.897 \pm 0.203$ | $0.505 \pm 0.046$ | $0.575 \pm 0.024$ | $0.269 \pm 0.011$ | $0.284 \pm 0.004$ |
| | | | 336 | $0.328 \pm 0.006$ | $0.568 \pm 0.047$ | $1.04 \pm 0.114$ | $0.609 \pm 0.025$ | $0.766 \pm 0.226$ | $0.289 \pm 0.007$ | $0.299 \pm 0.005$ |
| | | | 720 | $0.412 \pm 0.013$ | $0.695 \pm 0.006$ | $1.458 \pm 0.216$ | $0.66 \pm 0.048$ | $1.481 \pm 0.065$ | $0.327 \pm 0.003$ | $0.311 \pm 0.01$ |
| | PatchTST | w/ | 96 | $0.215 \pm 0.001$ | $0.399 \pm 0.0$ | $0.345 \pm 0.004$ | $0.368 \pm 0.007$ | $0.261 \pm 0.001$ | $0.276 \pm 0.001$ | $0.298 \pm 0.0$ |
| | | | 192 | $0.257 \pm 0.0$ | $0.431 \pm 0.001$ | $0.399 \pm 0.002$ | $0.387 \pm 0.004$ | $0.307 \pm 0.003$ | $0.283 \pm 0.0$ | $0.303 \pm 0.001$ |
| | | | 336 | $0.297 \pm 0.0$ | $0.460 \pm 0.004$ | $0.438 \pm 0.004$ | $0.409 \pm 0.001$ | $0.349 \pm 0.001$ | $0.298 \pm 0.001$ | $0.308 \pm 0.001$ |
| | | | 720 | $0.346 \pm 0.0$ | $0.5 \pm 0.002$ | $0.456 \pm 0.003$ | $0.443 \pm 0.001$ | $0.405 \pm 0.004$ | $0.331 \pm 0.0$ | $0.326 \pm 0.001$ |
| | | w/o | 96 | $0.234 \pm 0.002$ | $0.417 \pm 0.003$ | $0.392 \pm 0.034$ | $0.399 \pm 0.004$ | $0.305 \pm 0.005$ | $0.28 \pm 0.001$ | $0.306 \pm 0.001$ |
| | | | 192 | $0.273 \pm 0.003$ | $0.451 \pm 0.012$ | $0.472 \pm 0.012$ | $0.417 \pm 0.004$ | $0.391 \pm 0.056$ | $0.286 \pm 0.001$ | $0.308 \pm 0.0$ |
| | | | 336 | $0.316 \pm 0.009$ | $0.498 \pm 0.009$ | $0.52 \pm 0.019$ | $0.443 \pm 0.007$ | $0.424 \pm 0.012$ | $0.3 \pm 0.001$ | $0.315 \pm 0.001$ |
| | | | 720 | $0.378 \pm 0.013$ | $0.569 \pm 0.015$ | $0.698 \pm 0.032$ | $0.478 \pm 0.003$ | $0.5 \pm 0.008$ | $0.332 \pm 0.001$ | $0.334 \pm 0.001$ |
| | iTransformer | w | 96 | $0.214 \pm 0.001$ | $0.409 \pm 0.0$ | $0.35 \pm 0.001$ | $0.378 \pm 0.001$ | $0.271 \pm 0.0$ | $0.24 \pm 0.0$ | $0.268 \pm 0.0$ |
| | | | 192 | $0.257 \pm 001$ | $0.440 \pm 0.001$ | $0.399 \pm 0.0$ | $0.395 \pm 0.0$ | $0.311 \pm 0.001$ | $0.256 \pm 0.0$ | $0.278 \pm 0.001$ |
| | | | 336 | $0.299 \pm 0.001$ | $0.464 \pm 0.001$ | $0.433 \pm 0.001$ | $0.418 \pm 0.0$ | $0.352 \pm 0.001$ | $0.27 \pm 0.001$ | $0.284 \pm 0.0$ |
| | | | 720 | $0.35 \pm 0.0$ | $0.497 \pm 0.001$ | $0.447 \pm 0.001$ | $0.457 \pm 0.002$ | $0.406 \pm 0.001$ | $0.299 \pm 0.001$ | $0.3 \pm 0.001$ |
| | | w/o | 96 | $0.226 \pm 0.001$ | $0.433 \pm 0.002$ | $0.492 \pm 0.022$ | $0.409 \pm 0.004$ | $0.331 \pm 0.011$ | $0.252 \pm 0.001$ | $0.306 \pm 0.002$ |
| | | | 192 | $0.279 \pm 0.01$ | $0.469 \pm 0.001$ | $0.664 \pm 0.036$ | $0.437 \pm 0.004$ | $0.461 \pm 0.031$ | $0.266 \pm 0.001$ | $0.305 \pm 0.004$ |
| | | | 336 | $0.32 \pm 0.002$ | $0.512 \pm 0.004$ | $0.713 \pm 0.039$ | $0.471 \pm 0.007$ | $0.613 \pm 0.056$ | $0.288 \pm 0.0$ | $0.315 \pm 0.008$ |
| | | | 720 | $0.37 \pm 0.004$ | $0.565 \pm 0.003$ | $0.801 \pm 0.027$ | $0.509 \pm 0.009$ | $0.886 \pm 0.041$ | $0.322 \pm 0.006$ | $0.334 \pm 0.005$ |
| | TimeXer | w/ | 96 | $0.206 \pm 0.0$ | $0.404 \pm 0.001$ | $0.34 \pm 0.002$ | $0.36 \pm 0.002$ | $0.255 \pm 0.001$ | $0.243 \pm 0.002$ | $0.274 \pm 0.001$ |
| | | | 192 | $0.249 \pm 0.001$ | $0.439 \pm 0.002$ | $0.394 \pm 0.006$ | $0.385 \pm 0.001$ | $0.301 \pm 0.0$ | $0.256 \pm 0.001$ | $0.283 \pm 0.0$ |
| | | | 336 | $0.291 \pm 0.001$ | $0.462 \pm 0.01$ | $0.433 \pm 0.005$ | $0.407 \pm 0.0$ | $0.34 \pm 0.003$ | $0.274 \pm 0.002$ | $0.289 \pm 0.001$ |
| | | | 720 | $0.343 \pm 0.001$ | $0.487 \pm 0.018$ | $0.447 \pm 0.003$ | $0.441 \pm 0.001$ | $0.397 \pm 0.002$ | $0.303 \pm 0.005$ | $0.308 \pm 0.001$ |
| | | w/o | 96 | $0.232 \pm 0.002$ | $0.438 \pm 0.005$ | $0.756 \pm 0.078$ | $0.420 \pm 0.003$ | $0.412 \pm 0.013$ | $0.252 \pm 0.002$ | $0.32 \pm 0.003$ |
| | | | 192 | $0.277 \pm 0.002$ | $0.479 \pm 0.019$ | $0.864 \pm 0.027$ | $0.445 \pm 0.009$ | $0.548 \pm 0.066$ | $0.27 \pm 0.003$ | $0.321 \pm 0.003$ |
| | | | 336 | $0.334 \pm 0.007$ | $0.489 \pm 0.009$ | $0.906 \pm 0.016$ | $0.499 \pm 0.011$ | $0.683 \pm 0.082$ | $0.293 \pm 0.001$ | $0.322 \pm 0.002$ |
| | | | 720 | $0.399 \pm 0.002$ | $0.569 \pm 0.007$ | $1.036 \pm 0.054$ | $0.528 \pm 0.012$ | $1.219 \pm 0.067$ | $0.33 \pm 0.004$ | $0.343 \pm 0.004$ |

Table 5: The effect of Z-normalization on MSE of various models across benchmark datasets.

| Metric | Model | Z-Norm | Pred length | Weather | ETTh1 | ETTh2 | ETTm1 | ETTm2 | Electricity | Traffic |
|---|---|---|---|---|---|---|---|---|---|---|
| MSE | Crossformer | w/ | 96 | $0.154 \pm 0.003$ | $0.391 \pm 0.002$ | $0.315 \pm 0.006$ | $0.317 \pm 0.001$ | $0.171 \pm 0.001$ | $0.137 \pm 0.002$ | $0.421 \pm 0.004$ |
| | | | 192 | $0.203 \pm 0.001$ | $0.44 \pm 0.001$ | $0.396 \pm 0.006$ | $0.367 \pm 0.0$ | $0.248 \pm 0.002$ | $0.158 \pm 0.001$ | $0.44 \pm 0.006$ |
| | | | 336 | $0.265 \pm 0.002$ | $0.472 \pm 0.002$ | $0.456 \pm 0.014$ | $0.410 \pm 0.006$ | $0.312 \pm 0.005$ | $0.175 \pm 0.002$ | $0.477 \pm 0.007$ |
| | | | 720 | $0.343 \pm 0.003$ | $0.519 \pm 0.009$ | $0.46 \pm 0.003$ | $0.482 \pm 0.005$ | $0.413 \pm 0.002$ | $0.214 \pm 0.006$ | $0.546 \pm 0.011$ |
| | | w/o | 96 | $0.151 \pm 0.005$ | $0.398 \pm 0.009$ | $0.715 \pm 0.132$ | $0.373 \pm 0.025$ | $0.328 \pm 0.063$ | $0.147 \pm 0.001$ | $0.522 \pm 0.021$ |
| | | | 192 | $0.203 \pm 0.002$ | $0.454 \pm 0.02$ | $1.435 \pm 0.414$ | $0.486 \pm 0.063$ | $0.655 \pm 0.071$ | $0.169 \pm 0.007$ | $0.55 \pm 0.009$ |
| | | | 336 | $0.262 \pm 0.005$ | $0.621 \pm 0.077$ | $1.82 \pm 0.344$ | $0.677 \pm 0.045$ | $1.203 \pm 0.665$ | $0.197 \pm 0.009$ | $0.588 \pm 0.027$ |
| | | | 720 | $0.407 \pm 0.027$ | $0.815 \pm 0.023$ | $2.996 \pm 0.758$ | $0.747 \pm 0.086$ | $4.675 \pm 0.497$ | $0.241 \pm 0.003$ | $0.604 \pm 0.007$ |
| | PatchTST | w/ | 96 | $0.173 \pm 0.001$ | $0.380 \pm 0.001$ | $0.292 \pm 0.005$ | $0.329 \pm 0.012$ | $0.178 \pm 0.001$ | $0.184 \pm 0.0$ | $0.459 \pm 0.001$ |
| | | | 192 | $0.221 \pm 0.001$ | $0.431 \pm 0.002$ | $0.382 \pm 0.007$ | $0.366 \pm 0.005$ | $0.246 \pm 0.002$ | $0.191 \pm 0.0$ | $0.468 \pm 0.001$ |
| | | | 336 | $0.276 \pm 0.0$ | $0.482 \pm 0.013$ | $0.428 \pm 0.005$ | $0.397 \pm 0.001$ | $0.311 \pm 0.001$ | $0.207 \pm 0.0$ | $0.483 \pm 0.001$ |
| | | | 720 | $0.353 \pm 0.0$ | $0.519 \pm 0.002$ | $0.442 \pm 0.006$ | $0.457 \pm 0.001$ | $0.41 \pm 0.007$ | $0.249 \pm 0.001$ | $0.517 \pm 0.002$ |
| | | w/o | 96 | $0.169 \pm 0.001$ | $0.396 \pm 0.004$ | $0.364 \pm 0.052$ | $0.369 \pm 0.002$ | $0.215 \pm 0.01$ | $0.184 \pm 0.001$ | $0.569 \pm 0.001$ |
| | | | 192 | $0.21 \pm 0.001$ | $0.446 \pm 0.01$ | $0.488 \pm 0.016$ | $0.399 \pm 0.005$ | $0.344 \pm 0.072$ | $0.19 \pm 0.001$ | $0.581 \pm 0.0$ |
| | | | 336 | $0.262 \pm 0.005$ | $0.51 \pm 0.004$ | $0.574 \pm 0.045$ | $0.435 \pm 0.009$ | $0.412 \pm 0.023$ | $0.203 \pm 0.001$ | $0.597 \pm 0.001$ |
| | | | 720 | $0.342 \pm 0.01$ | $0.6 \pm 0.023$ | $0.972 \pm 0.104$ | $0.489 \pm 0.004$ | $0.545 \pm 0.021$ | $0.241 \pm 0.001$ | $0.632 \pm 0.001$ |
| | iTransformer | w/ | 96 | $0.175 \pm 0.001$ | $0.394 \pm 0.001$ | $0.3 \pm 0.001$ | $0.345 \pm 0.003$ | $0.185 \pm 0.001$ | $0.148 \pm 0.0$ | $0.43 \pm 0.003$ |
| | | | 192 | $0.224 \pm 0.001$ | $0.447 \pm 0.001$ | $0.379 \pm 0.0$ | $0.383 \pm 0.001$ | $0.25 \pm 0.001$ | $0.165 \pm 0.0$ | $0.452 \pm 0.003$ |
| | | | 336 | $0.281 \pm 0.001$ | $0.490 \pm 0.001$ | $0.423 \pm 0.002$ | $0.418 \pm 0.002$ | $0.316 \pm 0.001$ | $0.178 \pm 0.001$ | $0.473 \pm 0.001$ |
| | | | 720 | $0.36 \pm 0.001$ | $0.511 \pm 0.002$ | $0.43 \pm 0.003$ | $0.49 \pm 0.003$ | $0.412 \pm 0.002$ | $0.21 \pm 0.001$ | $0.523 \pm 0.005$ |
| | | w/o | 96 | $0.166 \pm 0.001$ | $0.418 \pm 0.001$ | $0.497 \pm 0.028$ | $0.374 \pm 0.005$ | $0.241 \pm 0.011$ | $0.153 \pm 0.0$ | $0.54 \pm 0.002$ |
| | | | 192 | $0.217 \pm 0.006$ | $0.476 \pm 0.002$ | $0.806 \pm 0.045$ | $0.420 \pm 0.005$ | $0.401 \pm 0.041$ | $0.169 \pm 0.001$ | $0.553 \pm 0.006$ |
| | | | 336 | $0.274 \pm 0.004$ | $0.542 \pm 0.011$ | $0.939 \pm 0.078$ | $0.469 \pm 0.009$ | $0.695 \pm 0.127$ | $0.185 \pm 0.001$ | $0.58 \pm 0.016$ |
| | | | 720 | $0.344 \pm 0.005$ | $0.596 \pm 0.003$ | $1.124 \pm 0.061$ | $0.537 \pm 0.01$ | $1.273 \pm 0.125$ | $0.221 \pm 0.006$ | $0.612 \pm 0.004$ |
| | TimeXer | w/ | 96 | $0.158 \pm 0.0$ | $0.385 \pm 0.002$ | $0.288 \pm 0.003$ | $0.324 \pm 0.003$ | $0.17 \pm 0.0$ | $0.141 \pm 0.001$ | $0.43 \pm 0.003$ |
| | | | 192 | $0.206 \pm 0.001$ | $0.435 \pm 0.003$ | $0.373 \pm 0.011$ | $0.364 \pm 0.001$ | $0.239 \pm 0.001$ | $0.158 \pm 0.0$ | $0.452 \pm 0.003$ |
| | | | 336 | $0.263 \pm 0.001$ | $0.492 \pm 0.013$ | $0.421 \pm 0.005$ | $0.396 \pm 0.001$ | $0.299 \pm 0.003$ | $0.176 \pm 0.002$ | $0.473 \pm 0.001$ |
| | | | 720 | $0.342 \pm 0.001$ | $0.506 \pm 0.03$ | $0.433 \pm 0.003$ | $0.453 \pm 0.002$ | $0.397 \pm 0.003$ | $0.207 \pm 0.007$ | $0.523 \pm 0.005$ |
| | | w/o | 96 | $0.163 \pm 0.002$ | $0.41 \pm 0.004$ | $1.132 \pm 0.152$ | $0.390 \pm 0.006$ | $0.370 \pm 0.027$ | $0.15 \pm 0.001$ | $0.573 \pm 0.003$ |
| | | | 192 | $0.211 \pm 0.001$ | $0.473 \pm 0.019$ | $1.559 \pm 0.089$ | $0.425 \pm 0.015$ | $0.639 \pm 0.141$ | $0.17 \pm 0.002$ | $0.595 \pm 0.006$ |
| | | | 336 | $0.274 \pm 0.004$ | $0.504 \pm 0.009$ | $1.618 \pm 0.084$ | $0.502 \pm 0.014$ | $0.999 \pm 0.213$ | $0.194 \pm 0.002$ | $0.604 \pm 0.004$ |
| | | | 720 | $0.379 \pm 0.006$ | $0.598 \pm 0.014$ | $1.912 \pm 0.14$ | $0.551 \pm 0.018$ | $2.616 \pm 0.208$ | $0.235 \pm 0.004$ | $0.643 \pm 0.006$ |

Table 6: The effect of Z-normalization on MAE of various models across synthetic datasets.

| Metric | Model | Z-Norm | Pred length | Synthetic ($\gamma = 0.95$) | | | | Synthetic ($\gamma = 0.5$) | | | |
|---|---|---|---|---|---|---|---|---|---|---|---|
| | | | | $\alpha = 0$ | $\alpha = 0.2$ | $\alpha = 0.4$ | $\alpha = 0.8$ | $\alpha = 0$ | $\alpha = 0.2$ | $\alpha = 0.4$ | $\alpha = 0.8$ |
| MAE | Crossformer | w/ | 96 | $0.424 \pm 0.0$ | $0.434 \pm 0.0$ | $0.434 \pm 0.0$ | $0.436 \pm 0.0$ | $0.789 \pm 0.0$ | $0.787 \pm 0.0$ | $0.801 \pm 0.0$ | $0.762 \pm 0.001$ |
| | | | 192 | $0.423 \pm 0.0$ | $0.431 \pm 0.0$ | $0.433 \pm 0.0$ | $0.449 \pm 0.0$ | $0.789 \pm 0.0$ | $0.788 \pm 0.0$ | $0.798 \pm 0.0$ | $0.759 \pm 0.0$ |
| | | | 336 | $0.423 \pm 0.0$ | $0.431 \pm 0.0$ | $0.432 \pm 0.0$ | $0.451 \pm 0.001$ | $0.786 \pm 0.0$ | $0.789 \pm 0.0$ | $0.797 \pm 0.0$ | $0.767 \pm 0.002$ |
| | | | 720 | $0.425 \pm 0.0$ | $0.435 \pm 0.0$ | $0.435 \pm 0.0$ | $0.472 \pm 0.004$ | $0.783 \pm 0.0$ | $0.79 \pm 0.0$ | $0.794 \pm 0.0$ | $0.772 \pm 0.001$ |
| | | w/o | 96 | $0.419 \pm 0.0$ | $0.427 \pm 0.0$ | $0.427 \pm 0.0$ | $0.421 \pm 0.0$ | $0.782 \pm 0.0$ | $0.784 \pm 0.0$ | $0.794 \pm 0.0$ | $0.748 \pm 0.001$ |
| | | | 192 | $0.418 \pm 0.0$ | $0.426 \pm 0.0$ | $0.427 \pm 0.0$ | $0.427 \pm 0.0$ | $0.781 \pm 0.0$ | $0.783 \pm 0.0$ | $0.792 \pm 0.0$ | $0.752 \pm 0.0$ |
| | | | 336 | $0.418 \pm 0.001$ | $0.426 \pm 0.0$ | $0.426 \pm 0.001$ | $0.428 \pm 0.0$ | $0.778 \pm 0.0$ | $0.782 \pm 0.0$ | $0.791 \pm 0.0$ | $0.749 \pm 0.0$ |
| | | | 720 | $0.42 \pm 0.0$ | $0.428 \pm 0.0$ | $0.429 \pm 0.0$ | $0.433 \pm 0.0$ | $0.774 \pm 0.0$ | $0.785 \pm 0.0$ | $0.787 \pm 0.0$ | $0.747 \pm 0.0$ |
| | PatchTST | w/ | 96 | $0.424 \pm 0.0$ | $0.433 \pm 0.0$ | $0.433 \pm 0.0$ | $0.438 \pm 0.0$ | $0.794 \pm 0.0$ | $0.789 \pm 0.0$ | $0.8 \pm 0.0$ | $0.769 \pm 0.001$ |
| | | | 192 | $0.422 \pm 0.0$ | $0.43 \pm 0.0$ | $0.433 \pm 0.0$ | $0.452 \pm 0.0$ | $0.793 \pm 0.0$ | $0.791 \pm 0.0$ | $0.797 \pm 0.0$ | $0.763 \pm 0.0$ |
| | | | 336 | $0.421 \pm 0.0$ | $0.43 \pm 0.0$ | $0.431 \pm 0.0$ | $0.452 \pm 0.0$ | $0.789 \pm 0.0$ | $0.79 \pm 0.0$ | $0.797 \pm 0.0$ | $0.775 \pm 0.006$ |
| | | | 720 | $0.421 \pm 0.0$ | $0.432 \pm 0.0$ | $0.434 \pm 0.0$ | $0.47 \pm 0.0$ | $0.786 \pm 0.001$ | $0.791 \pm 0.0$ | $0.794 \pm 0.0$ | $0.782 \pm 0.008$ |
| | | w/o | 96 | $0.418 \pm 0.0$ | $0.425 \pm 0.0$ | $0.424 \pm 0.0$ | $0.422 \pm 0.0$ | $0.787 \pm 0.0$ | $0.787 \pm 0.0$ | $0.794 \pm 0.0$ | $0.757 \pm 0.001$ |
| | | | 192 | $0.416 \pm 0.0$ | $0.424 \pm 0.0$ | $0.425 \pm 0.0$ | $0.426 \pm 0.0$ | $0.784 \pm 0.0$ | $0.785 \pm 0.0$ | $0.793 \pm 0.0$ | $0.756 \pm 0.0$ |
| | | | 336 | $0.416 \pm 0.0$ | $0.424 \pm 0.0$ | $0.424 \pm 0.0$ | $0.429 \pm 0.0$ | $0.782 \pm 0.001$ | $0.783 \pm 0.0$ | $0.791 \pm 0.0$ | $0.756 \pm 0.0$ |
| | | | 720 | $0.416 \pm 0.0$ | $0.425 \pm 0.0$ | $0.427 \pm 0.0$ | $0.434 \pm 0.0$ | $0.777 \pm 0.0$ | $0.786 \pm 0.0$ | $0.788 \pm 0.0$ | $0.753 \pm 0.001$ |
| | iTransformer | w | 96 | $0.42 \pm 0.0$ | $0.43 \pm 0.0$ | $0.43 \pm 0.0$ | $0.448 \pm 0.001$ | $0.792 \pm 0.0$ | $0.787 \pm 0.0$ | $0.8 \pm 0.0$ | $0.761 \pm 0.0$ |
| | | | 192 | $0.418 \pm 0.0$ | $0.426 \pm 0.0$ | $0.43 \pm 0.001$ | $0.452 \pm 0.001$ | $0.79 \pm 0.0$ | $0.788 \pm 0.0$ | $0.798 \pm 0.0$ | $0.759 \pm 0.0$ |
| | | | 336 | $0.417 \pm 0.001$ | $0.426 \pm 0.001$ | $0.428 \pm 0.001$ | $0.455 \pm 0.003$ | $0.788 \pm 0.0$ | $0.789 \pm 0.0$ | $0.797 \pm 0.0$ | $0.766 \pm 0.001$ |
| | | | 720 | $0.418 \pm 0.0$ | $0.429 \pm 0.0$ | $0.43 \pm 0.0$ | $0.469 \pm 0.002$ | $0.786 \pm 0.001$ | $0.791 \pm 0.0$ | $0.795 \pm 0.0$ | $0.772 \pm 0.001$ |
| | | w/o | 96 | $0.415 \pm 0.0$ | $0.425 \pm 0.0$ | $0.423 \pm 0.0$ | $0.429 \pm 0.004$ | $0.785 \pm 0.0$ | $0.785 \pm 0.0$ | $0.794 \pm 0.0$ | $0.752 \pm 0.0$ |
| | | | 192 | $0.413 \pm 0.0$ | $0.422 \pm 0.0$ | $0.424 \pm 0.0$ | $0.432 \pm 0.001$ | $0.783 \pm 0.0$ | $0.783 \pm 0.0$ | $0.793 \pm 0.0$ | $0.752 \pm 0.0$ |
| | | | 336 | $0.413 \pm 0.001$ | $0.422 \pm 0.001$ | $0.422 \pm 0.0$ | $0.429 \pm 0.001$ | $0.781 \pm 0.0$ | $0.782 \pm 0.0$ | $0.791 \pm 0.0$ | $0.75 \pm 0.001$ |
| | | | 720 | $0.413 \pm 0.0$ | $0.423 \pm 0.0$ | $0.424 \pm 0.0$ | $0.427 \pm 0.0$ | $0.776 \pm 0.0$ | $0.786 \pm 0.0$ | $0.788 \pm 0.0$ | $0.749 \pm 0.0$ |
| | TimeXer | w/ | 96 | $0.423 \pm 0.0$ | $0.433 \pm 0.0$ | $0.433 \pm 0.0$ | $0.442 \pm 0.001$ | $0.79 \pm 0.0$ | $0.786 \pm 0.0$ | $0.798 \pm 0.0$ | $0.76 \pm 0.001$ |
| | | | 192 | $0.421 \pm 0.0$ | $0.43 \pm 0.0$ | $0.434 \pm 0.0$ | $0.452 \pm 0.0$ | $0.79 \pm 0.0$ | $0.788 \pm 0.0$ | $0.796 \pm 0.0$ | $0.759 \pm 0.002$ |
| | | | 336 | $0.42 \pm 0.0$ | $0.43 \pm 0.0$ | $0.432 \pm 0.0$ | $0.455 \pm 0.0$ | $0.788 \pm 0.0$ | $0.788 \pm 0.0$ | $0.795 \pm 0.0$ | $0.766 \pm 0.001$ |
| | | | 720 | $0.42 \pm 0.0$ | $0.432 \pm 0.0$ | $0.435 \pm 0.0$ | $0.471 \pm 0.002$ | $0.785 \pm 0.0$ | $0.79 \pm 0.0$ | $0.793 \pm 0.0$ | $0.773 \pm 0.0$ |
| | | w/o | 96 | $0.417 \pm 0.0$ | $0.426 \pm 0.0$ | $0.425 \pm 0.0$ | $0.425 \pm 0.001$ | $0.783 \pm 0.0$ | $0.784 \pm 0.0$ | $0.792 \pm 0.0$ | $0.748 \pm 0.0$ |
| | | | 192 | $0.415 \pm 0.0$ | $0.424 \pm 0.0$ | $0.426 \pm 0.0$ | $0.428 \pm 0.0$ | $0.782 \pm 0.0$ | $0.782 \pm 0.0$ | $0.791 \pm 0.0$ | $0.751 \pm 0.001$ |
| | | | 336 | $0.415 \pm 0.0$ | $0.424 \pm 0.0$ | $0.424 \pm 0.0$ | $0.43 \pm 0.0$ | $0.78 \pm 0.0$ | $0.781 \pm 0.0$ | $0.789 \pm 0.0$ | $0.75 \pm 0.0$ |
| | | | 720 | $0.416 \pm 0.0$ | $0.424 \pm 0.0$ | $0.427 \pm 0.0$ | $0.434 \pm 0.0$ | $0.776 \pm 0.0$ | $0.785 \pm 0.0$ | $0.786 \pm 0.0$ | $0.749 \pm 0.0$ |

Table 7: The effect of Z-normalization on MSE of various models across synthetic datasets.

| Metric | Model | Z-Norm | Pred length | Synthetic ($\gamma = 0.95$) | | | | Synthetic ($\gamma = 0.5$) | | | |
|---|---|---|---|---|---|---|---|---|---|---|---|
| | | | | $\alpha = 0$ | $\alpha = 0.2$ | $\alpha = 0.4$ | $\alpha = 0.8$ | $\alpha = 0$ | $\alpha = 0.2$ | $\alpha = 0.4$ | $\alpha = 0.8$ |
| MSE | Crossformer | w/ | 96 | $0.283 \pm 0.0$ | $0.293 \pm 0.0$ | $0.296 \pm 0.0$ | $0.297 \pm 0.0$ | $0.972 \pm 0.0$ | $0.974 \pm 0.0$ | $1.0 \pm 0.0$ | $0.912 \pm 0.002$ |
| | | | 192 | $0.281 \pm 0.0$ | $0.29 \pm 0.0$ | $0.295 \pm 0.0$ | $0.316 \pm 0.001$ | $0.97 \pm 0.0$ | $0.975 \pm 0.0$ | $0.995 \pm 0.0$ | $0.907 \pm 0.001$ |
| | | | 336 | $0.28 \pm 0.001$ | $0.291 \pm 0.0$ | $0.295 \pm 0.0$ | $0.319 \pm 0.001$ | $0.964 \pm 0.0$ | $0.975 \pm 0.0$ | $0.991 \pm 0.0$ | $0.928 \pm 0.006$ |
| | | | 720 | $0.281 \pm 0.001$ | $0.295 \pm 0.0$ | $0.299 \pm 0.0$ | $0.351 \pm 0.005$ | $0.957 \pm 0.001$ | $0.978 \pm 0.0$ | $0.983 \pm 0.001$ | $0.941 \pm 0.003$ |
| | | w/o | 96 | $0.276 \pm 0.0$ | $0.284 \pm 0.0$ | $0.288 \pm 0.0$ | $0.279 \pm 0.0$ | $0.953 \pm 0.0$ | $0.965 \pm 0.0$ | $0.984 \pm 0.0$ | $0.878 \pm 0.002$ |
| | | | 192 | $0.274 \pm 0.0$ | $0.284 \pm 0.0$ | $0.288 \pm 0.0$ | $0.287 \pm 0.0$ | $0.949 \pm 0.0$ | $0.96 \pm 0.0$ | $0.982 \pm 0.0$ | $0.889 \pm 0.0$ |
| | | | 336 | $0.274 \pm 0.001$ | $0.284 \pm 0.001$ | $0.287 \pm 0.001$ | $0.289 \pm 0.001$ | $0.945 \pm 0.0$ | $0.957 \pm 0.0$ | $0.977 \pm 0.0$ | $0.883 \pm 0.001$ |
| | | | 720 | $0.275 \pm 0.0$ | $0.287 \pm 0.0$ | $0.29 \pm 0.0$ | $0.297 \pm 0.0$ | $0.937 \pm 0.0$ | $0.963 \pm 0.0$ | $0.966 \pm 0.001$ | $0.881 \pm 0.0$ |
| | PatchTST | w/ | 96 | $0.283 \pm 0.0$ | $0.292 \pm 0.0$ | $0.293 \pm 0.0$ | $0.3 \pm 0.0$ | $0.982 \pm 0.0$ | $0.981 \pm 0.0$ | $0.999 \pm 0.0$ | $0.929 \pm 0.002$ |
| | | | 192 | $0.281 \pm 0.0$ | $0.289 \pm 0.0$ | $0.294 \pm 0.0$ | $0.32 \pm 0.001$ | $0.977 \pm 0.0$ | $0.983 \pm 0.0$ | $0.995 \pm 0.0$ | $0.918 \pm 0.001$ |
| | | | 336 | $0.279 \pm 0.0$ | $0.29 \pm 0.0$ | $0.293 \pm 0.0$ | $0.32 \pm 0.0$ | $0.97 \pm 0.001$ | $0.98 \pm 0.001$ | $0.99 \pm 0.001$ | $0.948 \pm 0.016$ |
| | | | 720 | $0.276 \pm 0.0$ | $0.292 \pm 0.0$ | $0.297 \pm 0.0$ | $0.347 \pm 0.001$ | $0.963 \pm 0.001$ | $0.981 \pm 0.001$ | $0.982 \pm 0.001$ | $0.966 \pm 0.021$ |
| | | w/o | 96 | $0.275 \pm 0.0$ | $0.281 \pm 0.0$ | $0.282 \pm 0.0$ | $0.28 \pm 0.0$ | $0.962 \pm 0.0$ | $0.972 \pm 0.0$ | $0.984 \pm 0.0$ | $0.9 \pm 0.001$ |
| | | | 192 | $0.273 \pm 0.0$ | $0.281 \pm 0.0$ | $0.284 \pm 0.0$ | $0.287 \pm 0.0$ | $0.955 \pm 0.0$ | $0.967 \pm 0.0$ | $0.983 \pm 0.0$ | $0.899 \pm 0.001$ |
| | | | 336 | $0.273 \pm 0.0$ | $0.281 \pm 0.0$ | $0.283 \pm 0.0$ | $0.29 \pm 0.0$ | $0.951 \pm 0.002$ | $0.962 \pm 0.001$ | $0.977 \pm 0.001$ | $0.9 \pm 0.001$ |
| | | | 720 | $0.271 \pm 0.0$ | $0.282 \pm 0.0$ | $0.286 \pm 0.0$ | $0.297 \pm 0.0$ | $0.944 \pm 0.001$ | $0.967 \pm 0.001$ | $0.967 \pm 0.001$ | $0.897 \pm 0.002$ |
| | iTransformer | w/ | 96 | $0.277 \pm 0.0$ | $0.288 \pm 0.0$ | $0.29 \pm 0.0$ | $0.317 \pm 0.002$ | $0.977 \pm 0.0$ | $0.976 \pm 0.0$ | $0.999 \pm 0.001$ | $0.911 \pm 0.001$ |
| | | | 192 | $0.275 \pm 0.0$ | $0.283 \pm 0.001$ | $0.29 \pm 0.001$ | $0.325 \pm 0.002$ | $0.973 \pm 0.0$ | $0.977 \pm 0.0$ | $0.996 \pm 0.0$ | $0.907 \pm 0.0$ |
| | | | 336 | $0.273 \pm 0.001$ | $0.284 \pm 0.001$ | $0.289 \pm 0.001$ | $0.328 \pm 0.004$ | $0.967 \pm 0.0$ | $0.977 \pm 0.001$ | $0.993 \pm 0.0$ | $0.925 \pm 0.004$ |
| | | | 720 | $0.272 \pm 0.0$ | $0.287 \pm 0.0$ | $0.292 \pm 0.0$ | $0.348 \pm 0.003$ | $0.964 \pm 0.001$ | $0.981 \pm 0.0$ | $0.985 \pm 0.0$ | $0.939 \pm 0.002$ |
| | | w/o | 96 | $0.275 \pm 0.0$ | $0.281 \pm 0.0$ | $0.281 \pm 0.0$ | $0.291 \pm 0.006$ | $0.958 \pm 0.0$ | $0.967 \pm 0.0$ | $0.983 \pm 0.0$ | $0.89 \pm 0.001$ |
| | | | 192 | $0.273 \pm 0.0$ | $0.278 \pm 0.0$ | $0.283 \pm 0.001$ | $0.295 \pm 0.001$ | $0.952 \pm 0.0$ | $0.962 \pm 0.0$ | $0.983 \pm 0.0$ | $0.889 \pm 0.001$ |
| | | | 336 | $0.273 \pm 0.0$ | $0.279 \pm 0.001$ | $0.282 \pm 0.001$ | $0.292 \pm 0.001$ | $0.949 \pm 0.0$ | $0.96 \pm 0.001$ | $0.978 \pm 0.0$ | $0.886 \pm 0.001$ |
| | | | 720 | $0.271 \pm 0.0$ | $0.279 \pm 0.0$ | $0.283 \pm 0.0$ | $0.29 \pm 0.0$ | $0.942 \pm 0.0$ | $0.966 \pm 0.0$ | $0.968 \pm 0.0$ | $0.886 \pm 0.0$ |
| | TimeXer | w/ | 96 | $0.282 \pm 0.0$ | $0.293 \pm 0.0$ | $0.294 \pm 0.0$ | $0.307 \pm 0.001$ | $0.973 \pm 0.0$ | $0.974 \pm 0.0$ | $0.996 \pm 0.001$ | $0.908 \pm 0.002$ |
| | | | 192 | $0.28 \pm 0.0$ | $0.289 \pm 0.0$ | $0.295 \pm 0.0$ | $0.323 \pm 0.001$ | $0.972 \pm 0.0$ | $0.976 \pm 0.0$ | $0.991 \pm 0.001$ | $0.907 \pm 0.005$ |
| | | | 336 | $0.278 \pm 0.0$ | $0.29 \pm 0.0$ | $0.294 \pm 0.0$ | $0.327 \pm 0.0$ | $0.966 \pm 0.0$ | $0.974 \pm 0.0$ | $0.988 \pm 0.001$ | $0.926 \pm 0.004$ |
| | | | 720 | $0.276 \pm 0.0$ | $0.292 \pm 0.0$ | $0.297 \pm 0.001$ | $0.35 \pm 0.003$ | $0.961 \pm 0.0$ | $0.977 \pm 0.0$ | $0.982 \pm 0.001$ | $0.941 \pm 0.001$ |
| | | w/o | 96 | $0.275 \pm 0.0$ | $0.282 \pm 0.0$ | $0.284 \pm 0.0$ | $0.285 \pm 0.002$ | $0.954 \pm 0.0$ | $0.965 \pm 0.0$ | $0.978 \pm 0.0$ | $0.88 \pm 0.0$ |
| | | | 192 | $0.273 \pm 0.0$ | $0.281 \pm 0.0$ | $0.285 \pm 0.0$ | $0.289 \pm 0.0$ | $0.951 \pm 0.0$ | $0.96 \pm 0.0$ | $0.978 \pm 0.0$ | $0.887 \pm 0.004$ |
| | | | 336 | $0.273 \pm 0.0$ | $0.282 \pm 0.0$ | $0.284 \pm 0.0$ | $0.292 \pm 0.0$ | $0.948 \pm 0.0$ | $0.956 \pm 0.0$ | $0.972 \pm 0.0$ | $0.885 \pm 0.0$ |
| | | | 720 | $0.271 \pm 0.0$ | $0.281 \pm 0.0$ | $0.286 \pm 0.0$ | $0.297 \pm 0.0$ | $0.941 \pm 0.0$ | $0.963 \pm 0.0$ | $0.961 \pm 0.0$ | $0.887 \pm 0.0$ |

