# OpenReview forum: "A Closer Look at Transformers for Time Series Forecasting: Understanding Why They Work and Where They Struggle"
_ICML.cc/2025/Conference — ICML 2025 poster_

### Official Review · Reviewer_U2oa · 2025-02-26

**Overall Recommendation:** 2

**Summary:**

The paper investigates the effectiveness of Transformer-based models for time series forecasting, focusing on why simpler Transformers outperform more complex ones. These findings include that intra-variate dependencies dominate the performance of existing forecasting benchmarks, tokenization/channel independence are critical techniques to capture intra-variate patterns, and Z-score normalization significantly improves performance on non-stationary forecasting benchmarks. The authors delve into several representative Transformers on commonly adopted benchmarks and synthetic datasets.

**Claims And Evidence:**

> The assumption $H\left(\hat{\mathbf{x}}_j \mid \mathbf{x}_i, \mathbf{x}_{/ i}\right)=0$ (line 166)

This assumption is not right. The authors regard the model as a deterministic forecaster. However, the model is trained with probabilistic distributions, e.g., MSE specifies a Gaussian distribution.

> Point-wise Transformers underperform due to their weak capability of capturing intra-variate patterns (supported by Intra MI scores and synthetic experiments).

The claim that Transformers are less effective at capturing patterns within a single variate may not only stem from their tokenization (counterexamples such as recent pre-trained Transformers Time-MoE and Chronos). In this regard, the authors should also consider the sufficiency of training samples.

> Skip connections are crucial for capturing intra-variate dependencies but unsuitable to capture inter-variate patterns.

This conclusion can be highly dependent on the selected architecture of iTransformer. Broader validation on more types of Transformers (e.g., Transformers in Table 1) is needed.

**Essential References Not Discussed:**

The aforementioned works should also be cited.

**Experimental Designs Or Analyses:**

See Claims And Evidence.

**Methods And Evaluation Criteria:**

* The MI-based metrics assume deterministic outputs, which may not hold for probabilistic distribution for optimization.

**Other Comments Or Suggestions:**

Suggestion: (1) Include error bars in evaluations to assess variance. (2) Instead of assessing different types of Transformers, presenting respective ablation studies on disentangled components (e.g., tokenization, attention mechanism, channel independence, and Z-norm) can exclude their mutual influence. (3) It can be more helpful to provide showcases of synthetic datasets (under different configurations $\alpha$ $\gamma$) to illustrate the pipeline of Figure 1.

**Other Strengths And Weaknesses:**

* Strength: The metric of mutual information for time series forecasting is original.
* Weakness: (1) The findings presented by this paper need more comprehensive validation (e.g., evaluation on short-term forecasting datasets) (2) The paper can be further improved by providing insights into the solution to address these issue. (3) The description of data synthesis takes up too much space in the main text, while the conclusions drawn from the experiments are presented in a disorganized manner.

**Questions For Authors:**

The main conclusions are drawn by comparing the model's performance without multiple runs. Is the performance mutable and easily influenced by the training process? The authors need to provide the training configurations and error bars (at least in the appendix).

**Relation To Broader Scientific Literature:**

Several of the authors' findings in this paper have been mentioned in previous works:

* About the influence of Channel Independence: Rethinking Channel Dependence for Multivariate Time Series Forecasting: Learning from Leading Indicators. ICLR 2024.
* Z-Norm addresses distributional shift: Reversible Instance Normalization for Accurate Time-Series Forecasting against Distribution Shift. ICLR 2022.

**Theoretical Claims:**

See above.

---

> ### Author Rebuttal · Authors · 2025-03-28
>
> Thank you very much for your comments. We truly appreciate your effort and time for reviewing our work.
>
> *Response to the comment challenging the assumption that the model's output is deterministic:*
>
> &nbsp;&nbsp;&nbsp;&nbsp;    While MSE is linked to the assumption of Gaussian noise during training, it does not make the model's predictions stochastic. In this case, the model trained with MSE produces a single, fixed output for a given input. To obtain truly probabilistic or stochastic outputs, one would need to model and sample from the predictive distribution explicitly, such as through Bayesian methods or by estimating parameters of a distribution and sampling accordingly.
>
> *Response to the comment regarding pre-trained Transformers such as Time-MoE and Chronos:*
>
> &nbsp;&nbsp;&nbsp;&nbsp;    As stated at the end of Sec. Related Work, our study focuses on lightweight transformer architectures that are designed to be trained from scratch for an individual dataset. We did not include pre-trained models as they typically follow different training protocols and involve more complex architectures, making fair and transparent comparisons difficult. Additionally, Time-MoE uses a decoder-only architecture, which is fundamentally different from the encoder-based structures used in the lightweight models. Chronos, on the other hand, tokenizes time series into discrete bins through simple scaling and quantization, a strategy that differs significantly from the point-wise, patch-wise, and variate-wise tokenization. We appreciate the suggestion and will include references in the revision.
>
> *Response to the comment regarding broader validation of the importance of skip-connection:*
>
> &nbsp;&nbsp;&nbsp;&nbsp;    We conducted the skip-connection ablation study to explore the following questions (Line 327): “Why do transformers with basic attention mechanisms perform well in time series forecasting? Which components of the basic transformer architecture contribute most to this success?” Our focus is specifically on **basic and effective** transformer architectures. iTransformer meets both criteria, and we consider its structure representative for understanding key design components in transformer-based models. We analyzed skip connections in iTransformer because we hypothesized that they play a key role in learning intra-variate patterns, especially within an architecture designed to support inter-variate attention. Moreover, in models using point-wise tokens, skip connections primarily enhance inter-variate patterns, which differs fundamentally from the goal of this ablation study.
>
>
> *Response to comments regarding additional references.*
>
> &nbsp;&nbsp;&nbsp;&nbsp;    Thank you for your suggestions. While we agree that the suggested papers are relevant to the broader topic, we believe that the specific findings of our work are **not** addressed in them.
>
> &nbsp;&nbsp;&nbsp;&nbsp;    [1] Rethinking Channel Dependence... ICLR 2024
>
> &nbsp;&nbsp;&nbsp;&nbsp;    [1] proposes LIFT, a plugin method designed to identify and utilize locally stationary lead-lag relationships between variates to improve forecasting performance. While it focuses on modeling lead-lag dependencies, our work evaluated the effectiveness of various models in capturing general inter-variate relationships. The mutual information-based scores we propose are model-agnostic and capable of assessing inter-variate dependencies without being limited to specific relationship types.
>
> &nbsp;&nbsp;&nbsp;&nbsp;    [2] Reversible Instance Normalization... ICLR 2022.
>
> &nbsp;&nbsp;&nbsp;&nbsp;    Reversible Instance Normalization (RevIN), a method similar to Z-score normalization, was designed to address distributional shifts in time series data. A key difference is that RevIN incorporates learnable parameters within its normalization process, whereas the Z-score normalization examined in our study is the standard, non-learnable version. Moreover, our findings **contradict** the suggestion that "Z-Norm addresses distributional shift." Specifically, we observed that Z-score normalization degraded model performance on synthetic datasets which are non-stationary with monotonic trends (Table 4).
>
> *Response to other suggestions and questions:*
>
> &nbsp;&nbsp;&nbsp;&nbsp;    1. "...include error bars", "...without multiple runs", "...provide training configurations".
>
> &nbsp;&nbsp;&nbsp;&nbsp;    As stated in Sec. Experiments (Line 216 - 219): "All the experimental results in this work are averaged over 3 runs with different random seeds. Standard deviation of the results are provided in the Appendix." Both Fig. 3 and 4 include error bars. We will add hyperprameter configurations in the appendix in the revised manuscript.
>
> &nbsp;&nbsp;&nbsp;&nbsp;    2. "...provide showcases of synthetic datasets".
>
> &nbsp;&nbsp;&nbsp;&nbsp;    We will add visualization of synthetic datasets in the revised manuscript.

---

### Official Review · Reviewer_jMHQ · 2025-03-06

**Overall Recommendation:** 3

**Summary:**

The paper focuses on a recently heated topic and an important topic in time series forecasting -- which is to conduct further evaluation on the previously proposed models, not only on how a model can improve the forecasting performance, but also consider the performance changes are potentially related to the characteristics of the datasets. While the paper does not introduce a novel model architecture, it provides valuable insights into the generalization ability of Transformer-based models beyond widely used benchmark datasets (lots of them are poorly designed, see **Other Strengths And Weaknesses**).

**Claims And Evidence:**

See limitation with the **experimental designs** and **questions** to better support the claims.

**Essential References Not Discussed:**

There has been work discussing how data set characteristics could influence the model's predictive capacity [1]. The traditional methods have also included a lot of modelling / preprocessing designs for different data sets (eg. ARIMA with differencing for non-stationary data sets). Such modelling design should also be considered with modern time series forecasting in transformer models, and it would be of greater impact of this paper to provide a more comprehensive guidance towards how data characteristics can guide the development / choice of deep learning models.

- [1]: Forecast Evaluation for Data Scientists: Common Pitfalls and Best  Practices

**Experimental Designs Or Analyses:**

The experiments primarily rely on benchmark datasets that have already been used to evaluate the selected Transformer models. From this perspective, this paper is mainly a reproduction of the previous models on the already-tested data sets. However, these benchmark datasets alone are insufficient to draw a comprehensive understanding of how Transformers perform in time series forecasting (see Weaknesses). It would be more valuable to include additional high-quality datasets, particularly from diverse domains such as weather forecasting (weather bench etc.), power price prediction (see power price [1] used in TimeXer[2]), and other datasets that have long been established in specialized fields. This would help assess the models’ robustness and generalization across different types of time series data.

- [1] Forecasting day-ahead electricity prices: A review of state-of-the-art algorithms, best practices and an open-access benchmark
- [2] TimeXer: Empowering Transformers for Time Series Forecasting with Exogenous Variables

**Methods And Evaluation Criteria:**

The evaluation is conducted by averaging across the forecasting horizon, but a more comprehensive approach would be to assess also the model’s predictability for each forecasting horizon independently, as recommended in time series literature [1,2]. This is especially important for a paper claiming to reduce the error accumulation over the forecasting horizon.

[1] Another look at measures of forecast accuracy.
[2] Forecasting: Principles and Practice. Chapter 3.

**Other Comments Or Suggestions:**

see **questions for authors**

**Other Strengths And Weaknesses:**

**Data set:**
It has been argued that current benchmark datasets lack general predictive power (see: https://cbergmeir.com/talks/neurips2024/). For instance, the weather dataset requires a longer training time span (like PanguWeather[1], Aurora[2], ClimaX[3] etc.) and more external indicators and that the electricity load in the ETT dataset is also closely related to the weather, which potentially could benefit from the inclusion of weather indicators. The reliance on such low-quality datasets raises concerns about the generalisability of the conclusions drawn in this paper to real-life scenarios. A good thing about this paper, is that they also include a synthetic data set, which to some extent alleviates such concern.

- [1] Pangu-Weather: A 3D High-Resolution Model for Fast and Accurate Global Weather Forecast
- [2] Aurora: A Foundation Model of the Atmosphere
- [3] ClimaX: A foundation model for weather and climate

**Questions For Authors:**

**Synthetic data**
The use of synthetic datasets is a promising approach to explore how dataset characteristics affect model performance. However, several aspects of the dataset design would benefit from further clarification:
- **Limited Complexity**: The current design includes only two variates, which may not adequately reflect the complexity of real-world multivariate time series that often involve higher dimensionality and more intricate interactions. Could the authors clarify on the rationale behind this choice and whether extending the dataset to more variates was considered?
- **Simplified Dependency Structure**: The dependency parameter $\alpha$ appears to introduce only linear dependencies between the variates. This design may not sufficiently represent the diversity of relationships observed in real-world data, where dependencies are often non-linear, lagged, or state-dependent. Could the authors clarify whether non-linear or dynamic dependencies were considered, and how the dataset design could be extended to better capture such patterns?

**Relation To Broader Scientific Literature:**

see *Essential References*

**Theoretical Claims:**

See limitation with the **experimental designs** and **questions** to better support the claims.

---

> ### Author Rebuttal · Authors · 2025-03-27
>
> Thank you very much for your thoughtful comments. We truly appreciate your effort and time for reviewing our work.
>
> *Response to comments regarding additional metrics for evaluating forecasting accuracy:*
>
> &nbsp;&nbsp;&nbsp;&nbsp;    In this work, our primary focus is on understanding how different token representations within the attention mechanism lead to significant variations in model performance. We report MAE and MSE because they are the most widely used evaluation criterion in the literature, allowing for easy comparison with the related works. We acknowledge that this is not sufficient to evaluate models comprehensively. This is also the reason we introduce several mutual information-based scores in this work to explore model behaviour in capturing interactions between variates. The broader question of how to design and apply appropriate evaluation metrics for time series forecasting is a substantial topic in itself, requiring holistic analysis of both datasets and model behaviour. We consider this a valuable direction for future research.
>
> *Response to comments regarding the inclusion of additional datasets:*
>
> &nbsp;&nbsp;&nbsp;&nbsp;    We fully agree that the benchmark datasets commonly used for evaluating time series forecasting models should be significantly expanded. This is one of the key points we aim to highlight through our work. In this study, we selected the most widely used datasets in the literature, as they have become a standard reference for comparing model performance. However, based on our analysis, these benchmarks are limited in scope and do not sufficiently reflect the diversity of real-world forecasting challenges. To address this, we present results not only on these standard datasets but also on eight synthetic datasets and two real-world healthcare datasets, which help reveal the limitations of relying solely on several homogeneous benchmarks. We appreciate the suggestion and will examine the recommended datasets in our experimental settings.
>
> *Response to comments regarding essential references:*
>
> &nbsp;&nbsp;&nbsp;&nbsp;    Thank you for the valuable suggestion. We agree that considering dataset characteristics and appropriate modelling strategies—like traditional methods such as ARIMA—can greatly benefit modern time series forecasting. We see this as an important direction for future work, which should ideally be addressed alongside the analysis of dataset characteristics and the design of suitable evaluation metrics. We'll acknowledge this point in the revised manuscript.
>
> *Response to questions:*
>
> 1. Limited Complexity -- Why only two variates included in the synthetic data?
>
>     We used only two variates in the synthetic data to maintain clarity in analyzing inter-variate dependencies. We had considered to generate more variates, however, it made the analysis of results much less clear. For instance, introducing a third variate would require defining its dependency on the other two, resulting in two additional hyperparameters. Moreover, since the second variate is already dependent on the first, the dependency of the third variate would effectively be influenced by three parameters. This added complexity would make the analysis less transparent and interpretable.
>
> 2. Simplified Dependency Structure -- Why only linear dependencies between variates in the synthetic data?
>
>     Similarly, linear dependencies are straightforward to interpret and analyze. We had considered increasing the complexity of the dependencies if the models demonstrated strong performance in capturing linear relationships. However, even with these simple linear dependencies, we observed that patch-wise and variate-wise models still rely heavily on intra-variate dependency except in cases where the inter-variate dependency is extremely high (e.g. $\alpha = 0.8$). Given this, we did not see a clear benefit in introducing more complex dependencies at this stage.

---

### Official Review · Reviewer_iw3r · 2025-03-13

**Overall Recommendation:** 4

**Summary:**

There has been many proposals of transformer architectures for time series forecasting, and some of them are simpler some are more sophisticated, some work well, and some struggle. This work, examines why some of them work better and some struggle: In doing so, it uses an existing classification (Wang et al 2024), point wise, patch-wise and variate-wise tokens, and uses  a representative of each one, and introduce mutual information metric and additional synthetic datasets to test them. They show that intravariate dependencies are the primary source of contribution to prediction performance, while intervariate dependencies play rather a minor part.  Moreover,  Z-score normalisation and skip connections play a crucial role, in their ablation studies.  They validate these insights through synthetic and real world data sets.

##UPDATE AFTER REBUTTAL##

I maintain my initial positive score after the rebuttal, thanks to the quality of the work, and well-addressing of my questions by the authors.

**Claims And Evidence:**

In general claims and evidences are in line and are very well. There are some claims that are hard to justify in the presented tables:

Claim 1: "removing skip connection notably degrades the performance across most datasets, except for synthetic datasets with high inter- variate dependencies (α = 0.8) or low autocorrelation (γ = 0.5). "

The difference seems to minuscule, or does not exist: so mainly on the ones with the no interrelation (alpha = 0).  Moreover low autocorrelation does not seem to be degraded at all.


Claim 2:  "Replacing variate independent decoder with with a variant dependent decoder  improves performance on synthetic datasets with high inter-variate dependencies (α = 0.8) or low auto- correlation (γ = 0.5)."

The results are either so minuscule or non-existent that such bold claim that follows is difficult to make.

**Essential References Not Discussed:**

Temporal Fusion Transformer (Lim, Bryan, et al. "Temporal fusion transformers for interpretable multi-horizon time series forecasting." International Journal of Forecasting 37.4 (2021): 1748-1764.)

**Experimental Designs Or Analyses:**

I did check all of them. Only weak part is the claims regarding the synthetic datasets. So perhaps this is attributed to the way that the synthetic datasets are generated. Autoregression and interrelatedness are too simple, more free form function versions (inverse relations), nonlinear relations, sinusoidal seasonalities, or aperiodic cycles could justify the results better.

**Methods And Evaluation Criteria:**

Definitely makes sense. In general I like the mutual information tests, and the ablation studies ideas. In order to do that paper tries to characterise the data we are testing these systems, and this looks like the obvious viable option.

**Other Comments Or Suggestions:**

Include TFT in the analysis. Try to come up with a better synthetic data generation that can pronounce your claims stronger. (if they are really the case.)

Since mutual information metric is crucial it needs better explanation.

**Other Strengths And Weaknesses:**

Paper is very well written,  so the clarity is good. The work is original in the sense of methodology,  and quite significant, since there was quite many debates on the usefulness of transformers in forecasting with more traditional forecasting side was more skeptical. It can  definitely be influential in understanding transformers for forecasting.

 The paper also sheds light on the influential controversial paper showing some linear functions do better in forecasting (Zeng, A., Chen, M., Zhang, L., and Xu, Q. Are transform- ers effective for time series forecasting? In Proceed- ings of the AAAI conference on artificial intelligence, volume 37, pp. 11121–11128, 2023.)

The role of interdependencies between variates per data set, on the role of performance of certain architecture, is a message that makes utter sense.

Moreover, although kept a bit short, the finding of Avg Var Corr not being very crucial metric, and the effect of Z-normalisation  in TIHM and MINDEr datasets are important findings.

**Questions For Authors:**

1)Why did you not include TFT (perhaps the first model) in your results? (it is a highly-cited and commonly practiced successful model.)
2)What is the major role of MAX Mi?  Does it give us some extra information?
3) Could you explain \sigma ij, at the lowest level,  is it simply correlation?  (the N sample thing does not fully click in my mind, and the way they are selected.)

**Relation To Broader Scientific Literature:**

It is related to  causal discovery using transformer architectures e.g.,  [Kong, Lingbai, et al. "CausalFormer: An Interpretable Transformer for Temporal Causal Discovery." IEEE Transactions on Knowledge and Data Engineering (2024).]  since this very much depends on the transformer architecture taken into account.

**Theoretical Claims:**

There is no theoretical claim.

---

> ### Author Rebuttal · Authors · 2025-03-26
>
> Thank you very much for your encouraging comments. We truly appreciate your effort and time for reviewing our work.
>
> *Response to Claims And Evidence:*
>
> 1. In Table 3, removing the skip connection leads to a clear performance drop across all benchmark datasets, with particularly significant degradation on Electricity (MAE increases from 0.266 to 0.320) and Traffic (MAE increases from 0.283 to 0.591). On the synthetic datasets, this degradation becomes negligible while $\alpha$ increases under the condition of $\gamma$ = 0.95. In the manuscript, we stated that "removing the skip connection notably degrades performance across most datasets, **except** for synthetic datasets with high inter-variate dependencies ($\alpha$ = 0.8) or low autocorrelation ($\gamma$ = 0.5)," which aligns with your observation that "low autocorrelation does not seem to be degraded at all." We will revise this statement to improve its clarity in the revision.
>
> 2. Apologies for the confusion—this statement was incorrect. It should be: "It improves performance on synthetic datasets with high inter-variate dependencies ($\alpha$ = 0.8) under low autocorrelation ($\gamma$ = 0.5)." We will revise the corresponding claim to: "The variate-dependent decoder has the potential to enhance the model’s ability to capture inter-variate interactions, particularly in scenarios with strong dependencies between variates."
>
> *Response to broader literature and essential references:*
>
> &nbsp;&nbsp;&nbsp;&nbsp; Thank you for your valuable suggestions. We will add discussions of both papers in the Related Work section.
>
> *Response to questions:*
>
> 1. Why was TFT not included?
>
>     In this study, we focused on representative transformer architectures from the perspective of token representations within the attention mechanism. TFT, however, is a hybrid model that incorporates variable selection and LSTM-based encoders prior to applying attention. As a result, its token representations are less explicit and not as directly comparable to the models we selected. We agree that TFT is a relevant model and will include a discussion of it in the revised manuscript.
>
> 2. What is the major role of MAX MI?
>
>     We propose MAX MI as a measure of the mutual information captured by a model between the most strongly interacting variates. This metric helps assess whether a model is effectively learning inter-variate dependencies, particularly in cases where the number of variates is large and only a few exhibit strong interactions. For instance, in Figure 3, Crossformer achieves the highest MAX MI on the Traffic dataset (which has 862 variates), while its average inter-variate mutual information (Avg. Inter MI) remains lower than that of most other models.
>
> 3. Explanation of $\sigma_{ij}$?
>
>     $\sigma_{ij}$ estimates the extent to which changes in the prediction of variate $j$  are caused by changes in variate $i$. Unlike Pearson's correlation, it is more versatile as it captures both linear and non-linear relationships.
>     Mathematically, $\sigma_{ij}^2$ is the variance of the predictions of variate $j$ conditioned on inputs $\mathbf{x}$ where all variates are held fixed except for variate $i$. To estimate this conditional variance, the variation of variate $i$ is introduced through N different samples by augmenting original samples in the dataset.  Specifically, each original sample is augmented into N = 5 versions, differing only in the value of variate $i$:  one instance is set to zero, one retains the original value, and the remaining instances are generated by adding Gaussian noise of varying strengths to the original value. We will amend the explanation in the revised manuscript to improve the clarity of the paper.

---

### Official Review · Reviewer_npeJ · 2025-03-14

**Overall Recommendation:** 4

**Summary:**

The authors survey the literature on time-series forecasting with transformers, and divide previously published approaches into 3 categories.  Given multiple time-varying signals (variates), the signals can be chopped into tokens along the time axis, along the variate axis (one token per variate), or both.

The authors then analyze the performance of various transformer architectures on a variety of commonly used benchmark data sets.  They find that intra-variate dependencies in the data are much more predictive than inter-variate dependencies.  Point-wise models, which bundle all variates together into a single token, perform poorly because they fail to capture patterns within a single variate.

The authors also design a set of synthetic datasets, in which the inter-variate and intra-variate dependencies can be precisely controlled, to further back up their findings.

**Claims And Evidence:**

The authors do a good job on the literature survey, covering a variety of recently published architectures.  The experiments, using both benchmark datasets and custom synthetic datasets also seem well-designed.  The author's analysis is both thoughtful and detailed.

**Essential References Not Discussed:**

N/A

**Experimental Designs Or Analyses:**

The experiments seem to be well designed.

**Methods And Evaluation Criteria:**

The methods and evaluations make sense.

**Other Comments Or Suggestions:**

N/A

**Other Strengths And Weaknesses:**

N/A

**Questions For Authors:**

I do have one question.  I am not very familiar with time-series forecasting, but I was a bit confused after reading this paper.  The variate-wise models, like iTransformer, do not use attention to capture temporal (intra-variate) dependencies, and they still perform well.  But if attention is not being used to capture temporal dependencies, then why use a transformer for time-series modeling at all?

In normal language modeling, the whole point of attention is that it's very good at capturing long-range temporal dependencies.  If attention is not good at capturing such dependencies in time series data, then perhaps the transformer is not an appropriate architecture.  The authors further  reinforce this idea by pointing out that the skip connection, which bypasses attention, is crucial to model performance.  Am I missing something?

**Relation To Broader Scientific Literature:**

The literature survey, which compares the different architectures of various models, was particularly well done.

**Theoretical Claims:**

The only theory was in the equations for mutual-information.  I did not find any errors, but this is not my area of expertise.

---

> ### Author Rebuttal · Authors · 2025-03-26
>
> Thank you very much for your encouraging comments. We truly appreciate your effort and time for reviewing our work.
>
> *Response to questions:*
>
> 1. Why use transformers?
>
>     In previous work on developing new transformer architectures for time series forecasting, there has been little validation of whether the attention mechanism functions as intended. Although several studies aim to enhance transformers' ability to capture inter-variate dependencies, none have explicitly verified this in relation to the underlying data characteristics and model outcomes. This lack of validation motivated us to explore the issue in our study.
>
>     Our findings show that patch-wise and variate-wise transformers effectively capture intra-variate patterns, which remains a useful property in time series forecasting. Additionally, experiments with synthetic datasets demonstrate that certain transformer architectures—such as Crossformer—are capable of learning inter-variate patterns. This suggests that transformers can still be effective in scenarios where such patterns are important.  We hope that our findings can offer useful insights for designing new architectures or for more effectively applying transformers in related applications.
>
> 2. Is attention useful in iTransformer?
>
>     As we understand it, attention is still useful in iTransformer. For example, in our skip-connection ablation study, we observed that iTransformer performed better without skip connections on synthetic datasets with high inter-variate dependencies. This suggests that, in such scenarios, the attention mechanism alone is still capable of capturing meaningful patterns. Additionally, the skip connection in iTransformer is added to the attention output—not as a simple bypass, but rather as a means to reinforce the self-attention for each variate.

---

> > ### Comment · Reviewer_npeJ · 2025-04-04
> >
> > Thank you for the explanation.

---

### Decision · Program_Chairs · 2025-05-01

**Decision:**

Accept (poster)

**Comment:**

This paper studies the performance of various transformer architectures in time series forecasting, considering the impact of point-wise, patch-size, and 'variate'-wise tokenization, intra and inter-variate attention, and other components.

Results suggest that most time series benchmarks predominantly require modeling intra-variate patterns (rather than inter-variate patterns), and that normalization and skip-connections are crucial. These results are further supported by experiments on controlled synthetic datasets.

The paper received supportive scores from 3 out of 4 reviewers. Reviewers noted that the experiments were well designed, the paper was clear and it shed light on some surprising previous findings, e.g. 'Are transformers effective for time series forecasting? at AAAI.
Some weaknesses were identified that can be easily fixed, such as adding missing citations, and clarifying the limitations of the study (e.g., restricted to transformers trained on a single dataset, rather than so-called time series foundation models).

I agree with the majority of reviewers that the paper should be accepted. I encourage the authors to consider all feedback in their final camera-ready submission.